# Single-cell transcriptomics of the *Drosophila* wing disc reveals instructive epithelium-to-myoblast interactions

Nicholas J Everetts[1,2†], Melanie I Worley[1†], Riku Yasutomi[1], Nir Yosef[2*], Iswar K Hariharan[1*]

[1]Department of Molecular and Cell Biology, University of California, Berkeley, Berkeley, United States; [2]Department of Electrical Engineering & Computer Science, Center for Computational Biology, UC Berkeley, University of California, Berkeley, Berkeley, United States

**Abstract** In both vertebrates and invertebrates, generating a functional appendage requires interactions between ectoderm-derived epithelia and mesoderm-derived cells. To investigate such interactions, we used single-cell transcriptomics to generate a temporal cell atlas of the *Drosophila* wing disc from two developmental time points. Using these data, we visualized gene expression using a multilayered model of the wing disc and cataloged ligand–receptor pairs that could mediate signaling between epithelial cells and adult muscle precursors (AMPs). We found that localized expression of the fibroblast growth factor ligands, Thisbe and Pyramus, in the disc epithelium regulates the number and location of the AMPs. In addition, Hedgehog ligand from the epithelium activates a specific transcriptional program within adjacent AMP cells, defined by AMP-specific targets *Neurotactin* and *midline*, that is critical for proper formation of direct flight muscles. More generally, our annotated temporal cell atlas provides an organ-wide view of potential cell–cell interactions between epithelial and myogenic cells.

*For correspondence:
niryosef@berkeley.edu (NY);
ikh@berkeley.edu (IKH)

[†]These authors contributed equally to this work

Competing interests: The authors declare that no competing interests exist.

## Introduction

The development of multicellular eukaryotes gives rise to organs that are composed of cells of many types, typically derived from different germ layers such as the ectoderm and the mesoderm. There is increasing evidence that signaling during development between these distinct cell types plays an important role in ensuring the appropriate identity and number of cells in the fully formed adult organ (*Ribatti and Santoiemma, 2014*). A particularly well-studied example of such heterotypic interactions occurs during the development of the vertebrate limb, where signals are exchanged between the apical ectodermal ridge and the underlying mesoderm (*Delgado and Torres, 2017*).

While vertebrate limbs are relatively complex structures, the *Drosophila* wing-imaginal disc, the larval primordium of the adult wing and thorax, is ideally suited to the study of cell–cell interactions in the context of organ development because of its relative simplicity and amenability to genetic analysis (*Waddington, 1940*; *Cohen, 1993*; *Neto-Silva et al., 2009*). The wing-imaginal disc is composed of epithelial cells that form a sac-like structure (comprising the columnar cells of the disc proper and the squamous cells of the peripodial epithelium [PE]) and a population of adult muscle precursors (AMPs) that resides between the epithelial cells of the disc proper and the underlying basement membrane (*Figure 1A*). The epithelial portion of the disc derives from a primordium of approximately 30 cells from the embryonic ectoderm that are specified during embryogenesis (*Mandaravally Madhavan and Schneiderman, 1977*; *Worley et al., 2013*; *Requena et al., 2017*). The AMPs, originally referred to as adepithelial cells (*Poodry and Schneiderman, 1970*), represent a subset of cells from the embryonic mesoderm that generate the adult flight muscles (*Bate et al.,*

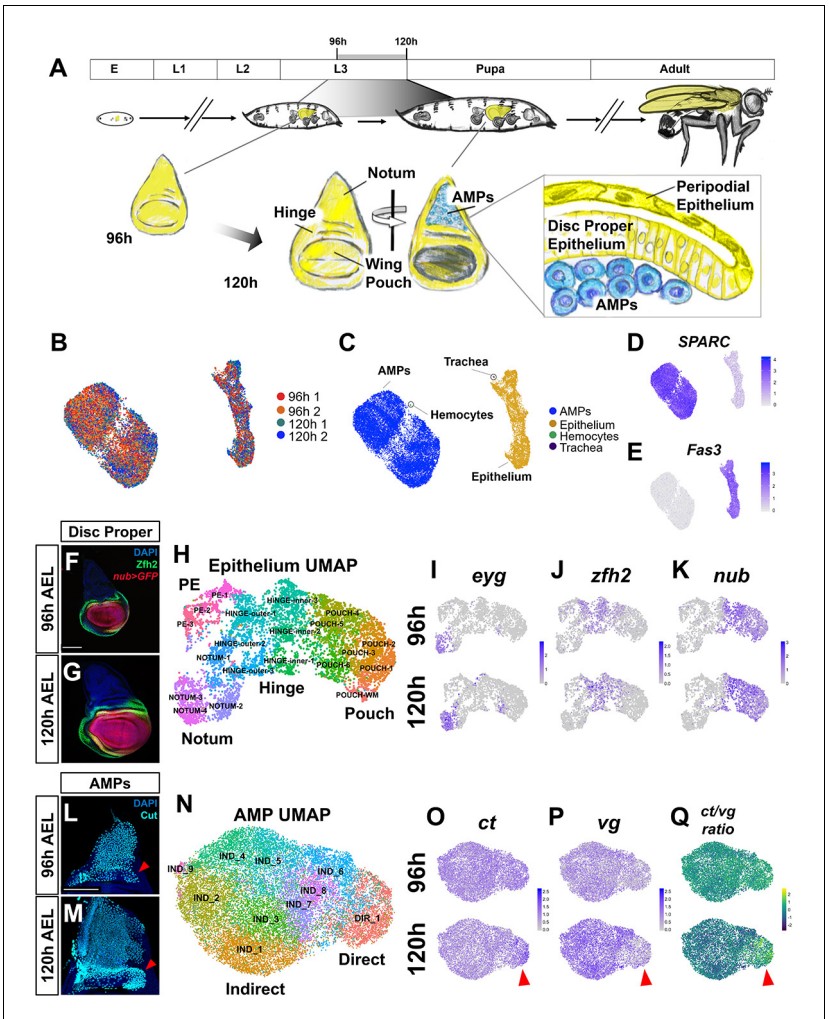

**Figure 1.** Temporal cell atlas of the developing wing-imaginal disc. (A) Timeline of *Drosophila* development: embryo (E), larval phases (L1–L3), pupa, and adult. Diagram of the wing-imaginal disc within the third larval instar (L3) from mid (96 hr) and late (120 hr) time points after egg lay (AEL). The epithelial cells of the wing disc become the adult wing blade, hinge, and the majority of the dorsal thorax (shown in yellow). The myoblasts associated with the basal surface of the disc proper epithelium are the adult muscle precursors (AMPs) (shown in blue), which generate the adult flight muscles. (B, C) UMAP of harmonized single-cell datasets with cells colored by batch (two time points with biological replicates) (B) and by major cell type (C). The AMPs and epithelial cells are distinguished by expression of *SPARC* (D) and *Fas3* (E), respectively. (F, G) Mid and late third instar wing-imaginal discs with the pouch marked by *nub-GAL4* driving the expression of GFP (red) and the hinge marked with anti-Zfh2 (green). DAPI (blue) stains all nuclei. (H) UMAP of harmonized epithelium cells, colored by unsupervised clustering identities and manually labeled by the expression of marker genes. (I–K) UMAPs with cells colored by their expression levels of *eyg* (I), *zfh2* (J), and *nub* (K) at both developmental time points. (L, M) Wing discs from 96 hr (L) and 120 hr (M) AEL, stained with anti-Cut to visualize the AMPs. Red arrowheads indicate location of direct AMPs, identifiable at 120 hr by higher anti-Cut staining and inferred by location at 96 hr. (N) UMAP of harmonized AMPs, colored by unsupervised clustering after cell cycle and cell sex stratification correction (see *Figure 1—figure supplements 4* and *5*). Cell clusters were classified as direct (DIR_1) or indirect AMPs (IND_1–9) based on marker gene expression (see *Figure 1—figure supplement 5E, F*). (O–Q) UMAPs separated by developmental time points showing the expression of canonical markers of direct and indirect AMPs, *ct* (O) and *vg* (P), and the ratio of these two genes within cells (Q). Red arrowheads highlight differential expression of these genes in the direct AMPs at 120 hr. Color scales for UMAPs correspond to normalized (by total unique molecular identifiers [UMIs]) counts on a natural-log scale. Microscopy scale bars = 100 μm.

The online version of this article includes the following figure supplement(s) for figure 1:

**Figure supplement 1.** Cell-type identification and temporal changes within major cell types.

**Figure supplement 2.** Temporal cell atlas of the wing disc epithelium.

*Figure 1 continued on next page*

*Figure 1 continued*

**Figure supplement 3.** Proximodistal axis stratifies the data more than anteroposterior axis.
**Figure supplement 4.** Correction of cell sex stratification within adult muscle precursor (AMP) scRNAseq data.
**Figure supplement 5.** Correction of cell cycle stratification within adult muscle precursor (AMP) scRNAseq data.
**Figure supplement 6.** Temporal changes within the direct and indirect adult muscle precursors (AMPs).

*1991*; *Fernandes et al., 1991*). The AMPs underlie the dorsal portion of the wing disc epithelium, the notum, which is the primordium of the dorsal thorax. During metamorphosis, these AMPs generate multiple muscle fibers that comprise the direct and indirect flight muscles (DFMs and IFMs, respectively) (reviewed by *Bothe and Baylies, 2016*; *Gunage et al., 2017*; *Laurichesse and Soler, 2020*).

The mechanisms that influence a seemingly uniform population of AMPs to generate different types of flight muscles, each composed of multiple distinct fibers, are not known. The AMPs are generated by an asymmetric division of a muscle founder cell during embryogenesis; one daughter cell becomes an AMP while the other generates precursors of larval muscles (*Bate et al., 1991*). In the second thoracic segment, dorsal clusters of AMPs, which express a segment-specific combination of Hox genes (*Roy and VijayRaghavan, 1997*), become associated with the wing disc, remain in the notum region, and proliferate via symmetric cell divisions. At the onset of the third larval instar (L3), the AMP cells switch to a pattern of asymmetric cell division as a result of Wingless (Wg; Wnt ligand) and Notch signaling interactions (*Gunage et al., 2014*). The precursors of the indirect and direct flight muscles can be distinguished by higher levels of expression of the transcriptional regulators Vestigial (Vg) or Cut (Ct), respectively (*Sudarsan et al., 2001*). The elevated Vg expression in the IFM precursors is maintained by expression of Wg ligand from the notum epithelium (*Sudarsan et al., 2001*).

Two large questions central to the mechanisms that regulate proliferation and cell fate specification in the AMPs, however, remain largely unanswered. First, what regulates the location and number of AMPs? It has been suggested that the notum acts as a 'dynamic niche' that both regulates the survival of AMPs and guides their specification (*Gunage et al., 2017*). However, the signals emanating from the epithelial cells to either regulate AMP numbers or maintain them in the notum region of the wing disc have not been identified. Second, it is known that the AMPs contribute to the formation of several complex muscle types: How are different AMP types specified? It has been proposed that extrinsic signals from the disc epithelium function during the larval stage to direct subsets of the AMPs to become precursors of specific types of muscles (*Gunage et al., 2017*). However, with the exception of Wg, such signals remain unidentified.

Single-cell transcriptomics provides a powerful strategy for mapping the cellular composition of developing organs (*Schier, 2020*), including vertebrate appendages (*Fabre et al., 2018*; *Cao et al., 2019*; *Feregrino et al., 2019*), and has been successfully utilized for characterizing the *Drosophila* wing disc at single snapshots during development (*Deng et al., 2019*; *Bageritz et al., 2019*; *Zappia et al., 2020*). Beyond cataloging cell types, transcriptome-scale analysis of single cells opens the way for a comprehensive evaluation of how interactions between cells may facilitate development (*Satija et al., 2015*; *Karaiskos et al., 2017*; *Deng et al., 2019*; *Bageritz et al., 2019*). Since spatial information is lost in most prevalent scRNAseq protocols, computational methods are used to infer it, usually based on the expression of landmark genes (*Satija et al., 2015*; *Karaiskos et al., 2017*; *Deng et al., 2019*; *Bageritz et al., 2019*). Complementary to spatial localization are approaches that have examined the expression of receptors, ligands, and downstream molecules to predict which cell subsets interact and by what mechanisms (*Vento-Tormo et al., 2018*; *Browaeys et al., 2020*).

The combination of these approaches, namely estimating the physical context of each cell and investigating the expression of extracellular cues such as receptor–ligand pairs, provides an unparalleled advantage for studying organ development. Here, we couple these two approaches to identify heterotypic interactions that are crucial for disc development, focusing on signaling between the disc epithelium and the AMPs. To this end, we collected single-cell RNA-sequencing (scRNAseq) data from two developmental time points, derived a comprehensive view of cell subsets and their spatial organization, and examined the expression of receptors and ligands. We show that

fibroblast growth factor (FGF) ligands emanating from the disc epithelium create an AMP niche that regulates AMP numbers and restricts them to the region of the notum. Furthermore, we find that Hedgehog ligand from the disc epithelium specifies a unique subpopulation of AMPs and identified novel Hh-target genes, *Neurotactin (Nrt)* and *midline* (*mid*), which are induced during the last phase of larval development. Beyond these examples, our annotated dataset provides a resource for spatiotemporal cellular composition in the developing wing disc and points to additional potential heterotypic interactions between epithelial cells and AMPs.

## Results

### Generation of a temporal cell atlas of the developing wing-imaginal disc

The wing disc is composed of multiple cell types, including the columnar cells of the disc proper, the squamous cells of the PE, and the mesoderm-derived AMPs (*Figure 1A*). In addition, the wing disc is in intimate contact with branches of the tracheal system and circulating blood cells called hemocytes. With the goal of generating a spatiotemporal atlas of the developing wing disc, we used single-cell RNA sequencing to collect transcriptional profiles of cells at mid and late third instar, which correspond to 96 and 120 hr after egg lay (AEL) (*Figure 1A*). Two biological replicates were obtained at each time point that, after filtering for low-quality cells, generated data from 6922 and 7091 cells in the 96 hr samples and 7453 and 5550 cells in the 120 hr samples. Harmonization of the different samples and dimensionality reduction was performed using scVI (*Lopez et al., 2018*). Clustering and differential expression analysis was done with the Seurat v3 R package (*Stuart et al., 2019*), and two-dimensional visualization of the data was performed with Uniform Manifold Approximation and Projection (UMAP) (*McInnes et al., 2018*) (see Materials and methods).

Our single-cell analysis identified four major cell types via known gene markers (*Figure 1B–E*, *Figure 1—figure supplement 1A–G*): the AMPs, the wing disc epithelial cells (disc proper and PE), and small numbers of tracheal cells and hemocytes. We observed expression of a similar set of marker genes in each of these cell types to those described by others (*Deng et al., 2019*; *Bageritz et al., 2019*; *Zappia et al., 2020*). Altogether, we recovered profiles for 19,885 AMPs, 7104 wing disc epithelial cells, 15 tracheal cells, and 12 hemocytes. Notably, our dataset shows an overrepresentation of AMPs, probably the result of our collagenase-based dissociation protocol (see Materials and methods) that dissociated AMPs far more effectively than epithelial cells. This enrichment of AMPs has enabled an especially detailed analysis of this cell type, which has previously received less attention. Unlike bulk RNA-sequencing approaches that would average changes in gene expression across multiple cell types, we were able to observe expression changes between 96 and 120 hr that occurred in both the epithelium and the AMPs together, as well as those changes that were confined to either the epithelial cells or the AMPs (*Figure 1—figure supplement 1H* and *Supplementary file 1*).

### Major transcriptional differences between epithelial cells reflect their proximodistal position

To search for the signals that might be exchanged between the epithelium and the AMPs, we first characterized the cell types within each of these populations separately. The wing disc epithelium is often divided into four broad domains – the notum, hinge, pouch, and PE – based both on morphology and gene expression patterns. The genes encoding the transcription factors *nubbin* (*nub*) and *Zn finger homeodomain 2* (*zfh2*) were used to define the pouch and hinge, respectively (*Zirin and Mann, 2007*; *Terriente et al., 2008*; *Ayala-Camargo et al., 2013*). Expression of these proteins is shown for wing discs at 96 and 120 hr AEL (*Figure 1F, G*). To characterize the epithelial compartment, we clustered the cells and then classified these clusters as originating from the larger domains of notum, hinge, pouch, and PE based on the expression patterns of marker genes (*Figure 1H–K*, *Figure 1—figure supplement 2A*). Many of these genes were already expressed in a domain-specific manner by 96 hr, suggesting that cells in the epithelium had already been partitioned into these domains (*Figure 1I–K*, *Figure 1—figure supplement 2B*).

The proximodistal axis was a primary feature in stratifying cells within our analysis, while the anteroposterior axis separated the data to a lesser degree. This is somewhat surprising because anterior

and posterior cells arise from distinct embryonic subpopulations (*Garcia-Bellido et al., 1973*; *Mandaravally Madhavan and Schneiderman, 1977*; *Worley et al., 2013*; *Requena et al., 2017*). We classified cells as anterior or posterior cells based on expression of compartment marker genes (e.g., *cubitus interruptus*, *hedgehog*, and *engrailed*) (*Figure 1—figure supplement 3A–E*) and found that the concise representation of the data in two dimensions (with UMAP) resulted in stratification of the cells based on the proximodistal axis first, with secondary stratification of cells based on anteroposterior identity (*Figure 1—figure supplement 3E*). Additional analysis indicated that there are more differentially expressed genes across the proximodistal axis than the anteroposterior axis (*Figure 1—figure supplement 3F–H*).

To investigate how cells changed over developmental time, we analyzed transcriptional changes that occurred within epithelium cell clusters between 96 and 120 hr AEL and found that 337 and 408 genes were significantly downregulated and upregulated, respectively, within at least one cluster (see Materials and methods; *Figure 1—figure supplement 2C*; *Supplementary file 2*). One example is *string* (*stg*) (*Edgar and O'Farrell, 1990*), which encodes a regulator of the cell cycle and is upregulated within the wing margin while being downregulated in other regions of the disc. Thus, even though the major cell types are established by 96 hr of development (*Figure 1I–K*, *Figure 1—figure supplement 2B*), we still find evidence of further pattern refinement via highly localized gene expression changes (*Figure 1—figure supplement 2C*).

## Cell-type identities among the AMPs are consolidated later than in the epithelium

Initial analysis of the AMPs showed a clear partition of the cells with respect to two primary features: cell sex (discs were collected from both male and female larvae) and cell cycle phase (*Figure 1—figure supplement 4A–G*, *Figure 1—figure supplement 5A–G*). We utilized scVI to suppress the effects of these covariates, which enabled us to obtain a clearer view of other aspects of AMP cell biology (see Materials and methods).

The AMPs are known to differentiate into either DFMs or IFMs of the adult fly (*Bate, 1993*; *Roy and VijayRaghavan, 1999*; *Sudarsan et al., 2001*). The precursors of these two populations can be identified by their location within the tissue and are canonically classified by their relative expression of two transcription factors, Vg and Ct, at the late third instar larval (L3) stage (corresponding to our 120 hr time point) (*Sudarsan et al., 2001*). The precursor cells of the IFMs are localized more dorsally (closer to the wing disc stalk) and display relatively high Vg and low Ct protein expression (*Sudarsan et al., 2001*). The DFM cell precursors are localized more ventrally (closer to the wing hinge) and are identifiable by little or no Vg and high Ct protein expression (*Sudarsan et al., 2001*). Ct expression at both time points is shown in *Figure 1L, M*. After unsupervised clustering (*Figure 1N*), we found that one cluster was characterized by high levels of *ct* and low levels of *vg*, and the remaining clusters displayed relatively elevated levels of *vg* and low levels of *ct* (*Figure 1O–Q*, *Figure 1—figure supplement 5E, F*). Based on this distinction, we classified cells as representing the direct or indirect AMPs (*Figure 1N*, *Figure 1—figure supplement 5F*), obtaining 17,604 indirect AMPs and 2281 direct AMPs.

Although the expression of *ct* and *vg* was less distinct at 96 hr, our analysis still classified cells from 96 hr into both direct and indirect populations. This prompted us to investigate if there were more subtle differences besides *ct* and *vg* at 96 hr that distinguished the direct and indirect AMPs. We identified a small number of genes that at both time points showed differential expression between direct and indirect AMPs (*Figure 1—figure supplement 6A*; *Supplementary file 3*), possibly indicative of pathways that are necessary both for the initial establishment and subsequent maintenance of AMP cell types. This included several predicted targets of Wg signaling, specifically *naked cuticle* (*nkd*) and *vg*, and the modulator of Notch signaling *fringe* (*fng*) (*Figure 1—figure supplement 6A, B*). However, the majority of differentially expressed genes that we identified were developmentally regulated. For these genes, differential expression between the direct and indirect AMPs was only observed at 120 hr. Thus, unlike the epithelium, the subdivision of the AMPs into direct and indirect pools (as assessed by differential expression of genes) mostly occurs only by 120 hr.

From 96 to 120 hr, we noticed the activation and refinement of expression of many genes previously implicated in axon guidance (*Figure 1—figure supplement 6A*). These include the receptor–ligand pairs *roundabout2/slit* and *Neurotactin/Amalgam* (*Kidd et al., 1999*; *Frémion et al., 2000*),

as well as the synaptic partner-matching genes *Ten-a* and *Ten-m* (*Hong et al., 2012*). We confirmed that Ten-m protein increases in expression from 96 to 120 hr by staining discs with anti-Ten-m antibody (*Figure 1—figure supplement 6C*). These observations suggest that pathways known to function in axon guidance could also function in myoblasts.

## A three-layered virtual wing disc and expression of ligand–receptor pairs predict cell–cell interactions

Since patterning of the epithelium precedes that of the AMPs, we looked for signals from the epithelium to the AMPs. First, to predict and visualize the location of gene expression within the wing disc, we generated a three-layered model, with gene expression levels inferred at different spatial positions within the AMP, disc proper, and PE layers. This enabled us to discover the expression of genes in regions of the disc epithelium that are closest to the AMPs as well as in the AMPs themselves. Second, we examined the expression of ligand–receptor pairs to identify those with complementary expression patterns between the disc epithelium and the AMPs.

To generate a three-layered transcriptomic map of the wing disc, we mapped our single-cell data to a reference model. The cells were mapped with the R package DistMap (*Karaiskos et al., 2017*), and the reference model was assembled from a manually curated set of gene expression patterns (*Figure 2A* and Materials and methods). This spatial mapping of cells was largely consistent with our manual cluster annotations based on marker genes (*Figure 2—figure supplement 1*). This included the four broad epithelial domains (notum, hinge, pouch, PE; *Figure 2B*) and more refined cell groups, such as the wing margin, posterior notum, outer pouch, and anterior hinge (*Figure 2C*). To test how well our virtual wing disc would predict novel gene expression patterns, we predicted the virtual in situ expression patterns of *grain* (*grn*) and *pou domain motif 3* (*pdm3*), neither of which were included in generating the disc model. The predicted patterns largely matched the expressions of transcriptional reporters for *grn* and *pdm3* within wing discs (*Figure 2D–G*), indicating that our virtual wing disc can successfully predict novel gene expression patterns.

To look for potential cell communication between the different cell layers of the wing disc, we examined the expression of ligand–receptor pairs within the major domains of the disc epithelium and AMPs (*Figure 2H*) (for details on receptor–ligand pairs examined, see Materials and methods). Interestingly, we observed high levels of expression of the genes encoding two FGF-family ligands, *thisbe* (*ths*) and *pyramus* (*pyr*) (*Stathopoulos et al., 2004*), in the notum region of the epithelium, whereas the gene encoding their receptor, *heartless* (*htl*) (*Beiman et al., 1996*), was specifically expressed in the AMPs. Similarly, the ligand *hedgehog* (*hh*) appears to be expressed only in the disc epithelium, while its receptor *patched* (*ptc*) and signal transducer *smoothened* (*smo*) are both expressed in the epithelium as expected, but also unexpectedly in the AMPs. Our detailed investigations of the FGF and Hedgehog pathways are presented in this study.

## FGF signaling from the epithelium creates a niche that regulates AMP number and localization

Ths and Pyr are the ligands for one of the *Drosophila* FGF signaling pathways, and both interact with the receptor Htl (*Stathopoulos et al., 2004*; *Figure 3A*). While *htl* is detected in nearly all of the AMPs, the three-layered disc map predicts that *ths* and *pyr* are expressed primarily in the epithelial cells of the notum, with considerable overlap (*Figure 3B–D*). Specifically, *ths* is localized to the most proximal region of the notum, while *pyr* has a broader expression pattern extending into the posterior hinge (compare *Figure 3C and D*). The expression patterns of *ths* and *htl* reporters were consistent with the expression predicted using our virtual wing disc (*Figure 3E, I*).

During embryogenesis, *htl*, *ths*, and *pyr* are known to influence mesoderm spreading along the embryonic ectoderm and formation of cardiac progenitor cells (*Beiman et al., 1996*; *Stathopoulos et al., 2004*; *Kadam et al., 2009*). Notably, in *htl* mutants and in *ths* and *pyr* double mutants, mesoderm cells are still present within the embryo, but they accumulate in multilayered arrangements instead of a monolayer along the ectoderm (as observed in wild-type embryos) (*Beiman et al., 1996*; *Stathopoulos et al., 2004*; *Kadam et al., 2009*). These observations suggest that FGF signaling might be primarily needed for proper mesoderm spreading rather than for cell proliferation and survival. By analogy, Ths and Pyr may regulate the localization of AMPs relative to the epithelium.

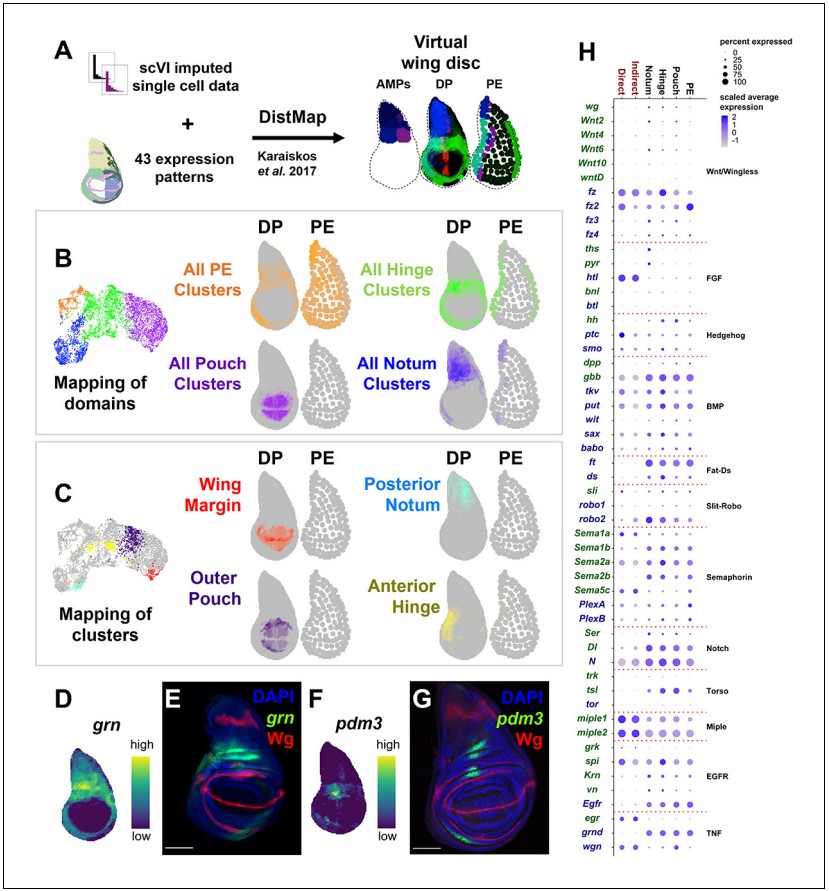

**Figure 2.** Spatial mapping of single-cell data to a virtual wing disc and analysis of receptor–ligand expression. (**A**) Schematic describing the creation of a three-layered virtual wing disc (adult muscle precursors [AMPs], disc proper [DP], and the peripodial epithelium [PE]) using DistMap (*Karaiskos et al., 2017*) (see Materials and methods). In contrast to the columnar cells of the DP, much of the PE is composed of squamous cells with flattened nuclei, and it is therefore represented as an outline that contains large dots. The virtual wing disc can be used to predict gene expression patterns (or virtual in situs), as shown for three example genes (*ptc* in red, *zfh2* in green, and *eyg* in blue). (**B**, **C**) Cells from the epithelial domains (**B**) and particular sub-regions (**C**) are shown both on the UMAP and mapped onto the DP and PE virtual wing disc. Stronger colors indicate higher predicted mapping; gray indicates low predicted mapping for cells (see *Figure 2—figure supplement 1A, B* for mappings of all cell clusters). (**D**, **F**) Predicted gene expression patterns of *grn* (**D**) and *pdm3* (**F**) in the epithelium disc proper, neither of which were used in building the virtual model. Yellow and dark purple correspond to high and low predicted expression, respectively. (**E**, **G**) Late third instar wing-imaginal disc with transcriptional reporters for the genes *grn* (**E**) and *pdm3* (**G**). Note the relative similarity between the predicted expression and transcriptional reporters. (**H**) Dot plot summarizing the expression of genes encoding receptors and ligands from pathways that were differentially expressed in at least one cell type. Dot size indicates the percent of cells that express the gene, and the dot color indicates the relative gene expression level within each of the cell groups. X-axis: cell groups. Disc epithelium cell types are in black font, and AMP cell types are in red font. Y-axis: genes are either in blue or green font depending on their annotation as encoding for a receptor or ligand, respectively. Microscopy scale bars = 100 µm.

The online version of this article includes the following figure supplement(s) for figure 2:

**Figure supplement 1.** Mapping of epithelium cell clusters to the virtual wing disc.

To examine the consequences of interfering with FGF signaling within the larval wing disc, we perturbed the expression of Pyr and Htl. To disrupt Pyr expression within the epithelial tissue, we utilized an *apterous* (*ap*) Gal4 driver (*ap-Gal4*) that expresses in the entire dorsal compartment of the disc proper, including all of the epithelial cells that overlie the AMPs (*Figure 3F, G*). Expressing an RNAi that targets *pyr* with this *ap-Gal4* driver resulted in a reduction in AMPs and an obvious

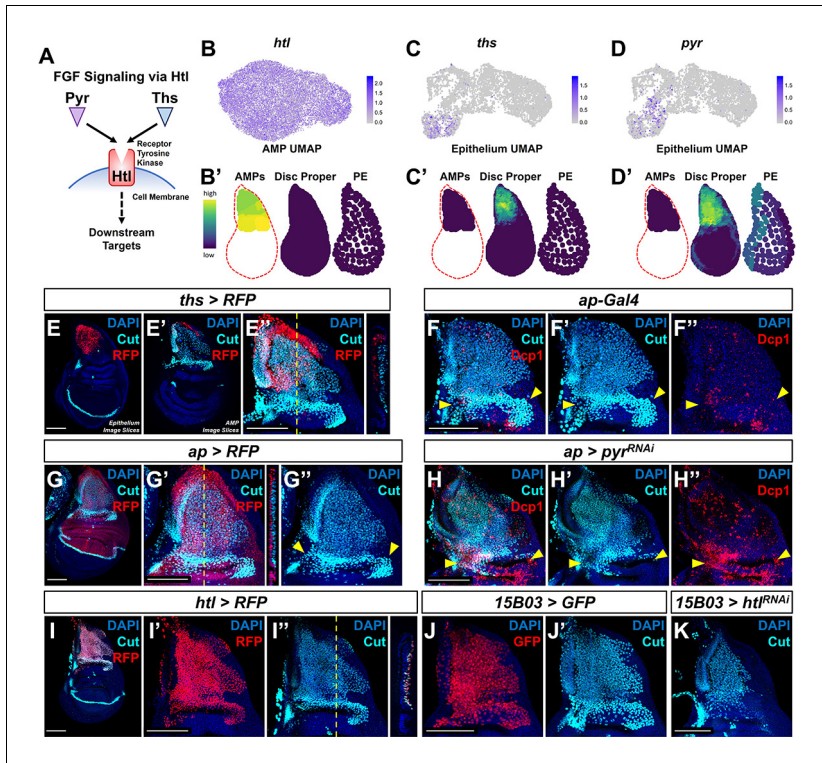

**Figure 3.** Fibroblast growth factor (FGF) signaling between adult muscle precursors (AMPs) and epithelium is critical for AMP viability and numbers. (**A**) FGF signaling pathway diagram. FGF ligands (Pyr and Ths) signal through the FGFR (Htl). (**B–D'**) Expression of *htl* (**B, B'**), *ths* (**C, C'**), and *pyr* (**D, D'**) in the single-cell data. (**B**), (**C**), and (**D**) show UMAPs of these genes, in either the AMPs (for *htl*) or disc epithelium (for *ths* and *pyr*). (**B'**), (**C'**), and (**D'**) show virtual disc map predictions for the expression of these genes in the AMPs, disc proper, and peripodial epithelium. (**E–E''**) *ths* expression domain (as shown by *ths-Gal4* driving the expression of RFP) in the wing disc. (**E**), (**E'**), and (**E''**) are max projections over image slices of the disc epithelium, images slices of the AMPs, and all image slices (both epithelium and AMPs), respectively. AMPs are visualized with anti-Cut (cyan). Orthogonal section (apical is left, basal is right) corresponds to the dashed yellow line in (**E''**). Note that *ths* reporter expression is specific to the notum epithelium and absent from the AMPs. (**F–H''**) Notum regions of wing discs with *ap-Gal4* transgene alone (**F–F''**) or *ap-Gal4* transgene driving expression of either >RFP (**G–G''**) or >pyr^RNAi (**H–H''**). AMPs are visualized by anti-Cut staining (cyan), shown in (**F, F', G–H'**); cell death is visualized by anti-Dcp1 staining (red), shown in (**F, F'', H, H''**). RFP (red fluorescent protein) expression shown in (**G, G''**) indicates that the domain of *ap-Gal4* covers the entire epithelium adjacent to the AMPs. Dashed yellow line in (**G'**) corresponds to the adjacent orthogonal section. Yellow arrowheads indicate expected location of direct AMPs. Note the loss of ventral- and posterior-localized AMPs following *pyr* knockdown, with increased anti-Dcp1 staining (**H–H''**). (**I–I''**) *htl* expression domain (as shown *htl-Gal4* driving the expression of RFP) in the wing disc. (**I'**) and (**I''**) show DAPI and either RFP or anti-Cut staining, respectively. Orthogonal section corresponds to the dashed yellow line in (**I''**) (contrast with the orthogonal section in **E''**). Note that the *htl* reporter is expressed by the AMPs, which are visualized by anti-Cut staining (cyan). (**J–K**) The notum regions from wing discs with AMP-specific *15B03-Gal4* driving the expression of >GFP (**J, J'**) and >htl^RNAi (**K**). AMPs visualized with anti-Cut. Note the reduction of AMPs, especially of the direct AMPs. UMAP color scales correspond to normalized counts on a natural-log scale. All notum images are max projections across image slices. Microscopy scale bars = 100 μm.

increase in apoptosis (visualized using anti-Dcp1), primarily observed in the more ventral- and posterior-localized AMPs (*Figure 3H*, compare with *Figure 3F*). A likely explanation for this result is that *pyr* knockdown within the wing disc restricts AMP survival to the Ths-expressing region of the dorsal notum as Pyr and Ths have been noted to have partially redundant functions (*Stathopoulos et al., 2004*; *Kadam et al., 2009*). Moreover, this result suggests that sufficient Ths cannot reach this region of the notum to compensate for the knockdown of Pyr at physiological levels of Ths expression. We next tested if the Ths and Pyr receptor Htl is required within the AMPs. Using the AMP-specific driver *15B03-Gal4* (*Figure 3J*), we expressed an RNAi for *htl* and observed an obvious decrease

in the number of AMPs following knockdown of FGF signal transduction (*Figure 3K*). Altogether, we conclude that FGF signaling between AMPs and the disc epithelium is necessary for proper AMP survival.

To determine if the location and level of FGF signaling controls the position and number of the AMPs, we assessed the effects of increasing the levels of FGF ligands and also expressing them ectopically. First, we used a *dpp-Gal4* driver that is expressed in a stripe of cells just anterior to the anterior-posterior compartment boundary of the epithelium, including in the notum (*Figure 4A*). Expression of either *pyr* or *ths* in this domain caused a massive increase in the number of AMPs, not only beneath the notum epithelium where AMPs are normally present, but throughout the entire *dpp-Gal4* expression domain, including underneath the wing pouch (*Figure 4B*, *Figure 4—figure supplement 1A*). AMPs were observed along the entire stripe of ectopic FGF expression, all the way to the ventral hinge and even extending on the ventral side to the peripodial epithelium. From this, we conclude that FGF signaling does not just increase AMP number, but can also induce AMP spreading beyond the epithelial notum.

We further investigated the role of FGF signaling in AMP migration by generating a separate patch of *pyr* expression, discontinuous from the domain of endogenous expression. To this end, we ectopically expressed *pyr* within the wing pouch using the TRiP-Overexpression VPR toolkit (*Lin et al., 2015*) with a *nub* driver. At the onset of *nub* expression, the pouch and the notum are separated by multiple cell diameters. We observed a large number of AMPs basal to the epithelium of the wing pouch (*Figure 4C, D*), suggesting that ectopic Pyr expression can even recruit AMPs from a distance of several cell diameters.

To ascertain whether the AMPs were capable of proliferation at ectopic locations, we assessed whether the cells were progressing through the cell cycle. AMPs throughout the A-P axis of the disc were found to be in both S- and M-phase of the cell cycle, as indicated by the incorporation of the thymidine analog EdU and staining for phospho-histone-H3 (PHH3) (*Figure 4E, F*, *Figure 4—figure supplement 1D, E*). This shows that when an FGF source is provided in the epithelium AMPs are capable of cell cycle entry and progression even in portions of the disc that are distant from the notum, consistent with the large increase in the number of AMPs. Thus, expression of Pyr and Ths in epithelial cells appears sufficient to both attract AMPs to that location and sustain their local proliferation.

To determine whether AMPs can be induced to migrate at different stages of larval development, we restricted the ectopic expression of FGF ligands to the later stages of larval development. The expression of *dpp>pyr* or *dpp>ths* was controlled by a temperature-sensitive Gal80, and when expression was initiated in mid-L3 (48 hr prior to dissection), AMPs were observed throughout the dorsal portion of the wing pouch (*Figure 4G*, *Figure 4—figure supplement 1B*). In contrast, when expression was initiated later in L3 (24 hr prior to dissection) AMPs were observed in the dorsal hinge but not the pouch (*Figure 4H*, *Figure 4—figure supplement 1C*). Taken together, these experiments indicate that the myoblasts found to be associated with other regions of the disc are derived from those underlying the notum. Additionally, these experiments show that AMPs can be induced to emigrate from the notum region even in the later stages of L3, suggesting that the localized expression of the FGF ligands in the notum both localizes AMPs to that region of the disc and also sustains their survival and proliferation (*Figure 4I*). Thus, FGF signaling effectively defines the AMP niche.

## Hh signaling regulates gene expression in a subset of posterior localized AMPs

Our analysis indicated that *patched* (*ptc*), which encodes the transmembrane receptor for the ligand Hh and is also a transcriptional target for Hh signaling, is expressed at low levels in most AMPs and at a much higher level in a subset of the direct AMPs (*Figure 5A*). Moreover, Hh-signaling pathway components *smoothened* (*smo*) and *cubitus interruptus* (*ci*) are expressed in most AMPs at uniform levels (*Figure 5B, C*). However, we detected negligible levels of *hh* transcripts within our AMP data, and furthermore did not detect expression of *hh-Gal4* within the AMPs (*Figure 5D*, *Figure 5—figure supplement 1B*). In contrast, *hh* transcripts were detected in approximately 32% of cells in the epithelium, roughly the size of the Hh-producing posterior compartment (*Figure 1—figure supplement 3B*). Together, these observations support the possibility that Hh from posterior cells of the disc

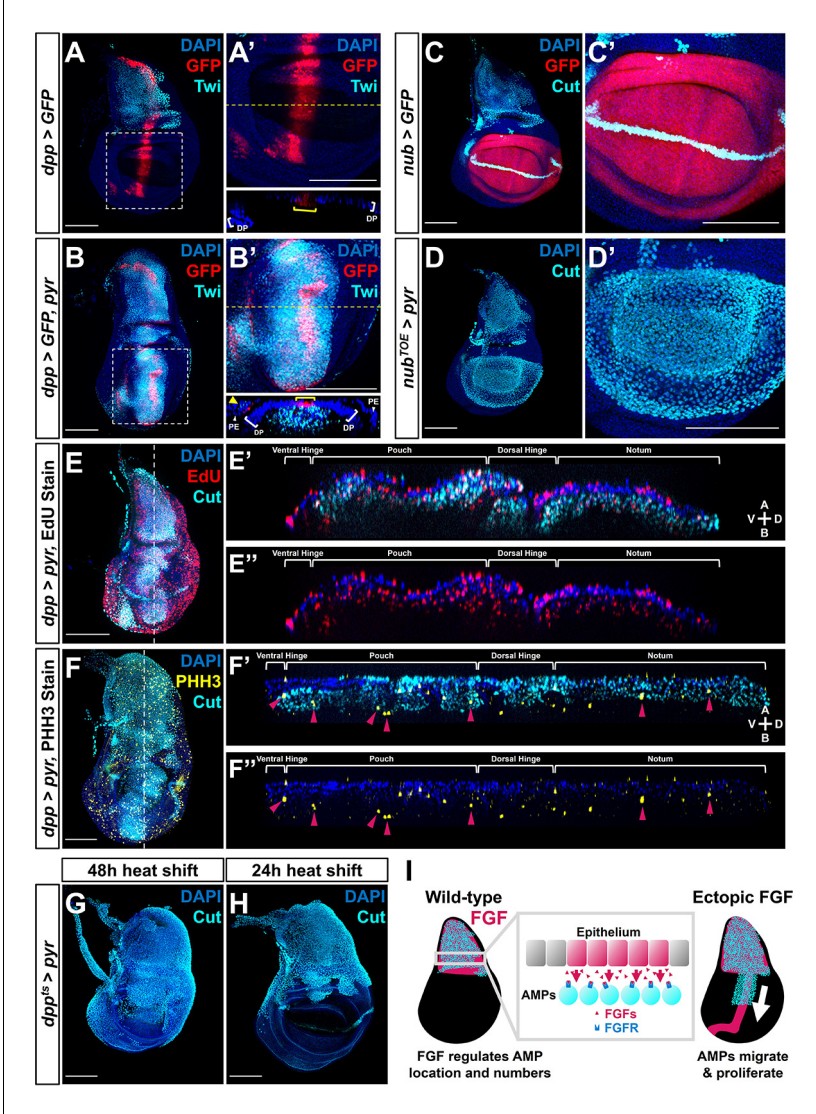

**Figure 4.** Fibroblast growth factor (FGF) from disc epithelium controls adult muscle precursor (AMP) localization. (A-B') Wing discs with *dpp-Gal4* driving the expression of >*GFP* alone (A, A') or >*GFP* together with >*pyr* (B, B'), stained with anti-Twi (cyan) to visualize the AMPs. (A') and (B') correspond to white dashed boxes in (A) and (B), respectively. Orthogonal sections (apical is top, basal is bottom, relative to disc proper) correspond to dashed yellow lines in (A') and (B'). Yellow brackets in orthogonal sections indicate *dpp-Gal4* expression in disc proper. Yellow arrowhead in the orthogonal of (B') indicates *dpp-Gal4* expression in the peripodial epithelium (PE), which recruits AMP expansion to the PE surface when expressing >*pyr*. (C–D') Wing discs with *nub-Gal4* driving the expression of >*GFP* (C, C') or >*dCas9VPR* (D, D'), the latter being used in conjunction with a guide RNA targeting an upstream sequence of the *pyr* transcriptional start site (*pyr TOE.GS00085*) to drive the overexpression of *pyr* in the wing pouch. Discs are stained with anti-Cut to visualize AMPs. Note that Cut is also expressed in the future wing margin of the disc epithelium (seen as a band through the wing pouch in C and C'). (E–F'') Discs overexpressing *pyr* via *dpp > pyr*, stained for Cut (cyan) and either EdU incorporation (red) (E–E'') or phosphohistone H3 (PHH3) (yellow) (F–F'') to assess the ability of AMPs to replicate DNA and undergo mitosis. Magenta arrowheads in (F') and (F'') highlight some AMPs that stain for PHH3. White brackets denote approximate domains of the disc proper. Note that ectopic AMPs stain for both EdU incorporation and PHH3, indicating that these cells are viable and proliferating outside of the endogenous AMP niche. A: apical; B: basal; V: ventral; D: dorsal, relative to disc proper. (G, H) Temperature-controlled expression of >*pyr* within the *dpp* domain, initiated at either mid L3 (G) or late L3 (H) (48 or 24 hr prior to pupariation, respectively). Note that even at these developmental stages we observed ectopic AMPs that appear to be emigrating ventrally from the notum. (I) Model for the effects of FGF overexpression on AMP growth. FGF/FGFR interactions between the disc epithelium and adjacent AMPs are necessary for AMP viability, and ectopic expression of FGF ligands induces emigration of AMPs from the notum to

*Figure 4 continued on next page*

*Figure 4 continued*

a domain that broadly matches the pattern of FGF ligand expression. All wing disc images are max projections across all image slices. Microscopy scale bars = 100 μm.

The online version of this article includes the following figure supplement(s) for figure 4:

**Figure supplement 1.** Ectopic Ths expression increases adult muscle precursor (AMP) number and migration.

epithelium activates Hh signaling in a subset of myoblasts, which respond by expressing higher levels of *ptc*.

We detected Ptc protein in a subpopulation of AMPs, localized primarily beneath the posterior compartment of the disc epithelium and extending approximately 20–40 μm into the region underlying the anterior compartment (*Figure 5E*, *Figure 5—figure supplement 1*). Consistent with our scRNAseq data, Ptc was observed mostly in a group of direct AMPs, but also in two additional smaller groups of cells that are located more dorsally among indirect AMPs. The proximity of these Ptc-expressing AMPs to Hh-secreting epithelial cells suggests that they are responding to the Hh secreted by these cells rather than a circulating pool of Hh that should be available to all AMPs. Indeed, a recent study suggests that Hh ligand from the epithelium is transported via cytonemes to nearby AMPs (*Hatori and Kornberg, 2020*).

To determine if Ptc expression in the AMPs closest to the Hh-producing epithelial cells is a result of Hh pathway activation, we reduced *smo* expression in all AMPs and observed that Ptc expression in AMPs was abolished (*Figure 5F*). This indicated that, as in the disc epithelium, Hh signal transduction within the AMPs is required to establish high Ptc expression in the posterior-localized AMPs. To address whether all AMPs are capable of this response, we expressed an activated form of the transcription factor Ci (Ci$^{3m}$), which is resistant to proteolytic cleavage (*Price and Kalderon, 1999*), in all AMPs and observed that this leads to an uniformly high level of Ptc protein expression (*Figure 5G*). Thus, all AMPs appear capable of responding to Hh, but during normal development, only AMPs with close proximity to the posterior compartment of the disc epithelium receive the signal. Neither *smo$^{RNAi}$*-knockdown nor *ci$^{3m}$* overexpression caused obvious changes in AMP numbers (compare *Figure 5E* with *Figure 5F, G*), suggesting that Hh signaling is not controlling AMP proliferation but instead is likely important for patterning.

To investigate a possible role of Hh signaling in AMP cell fate specification, we examined adult flies after genetic perturbations for flight muscle defects. During the pupal phase, the AMPs give rise to three distinct muscle fiber types within the adult thorax: dorsal longitudinal muscles (DLM), dorsoventral muscles (DVM), and DFMs (*Figure 5H*). While DLMs and DVMs are IFMs that generate the mechanical movement required for flight by compressing the thorax, the DFMs are responsible for flight steering by fine-tuning the position of the wing blades (reviewed by *Bate, 1993*). Both DLMs and DVMs are formed from indirect AMPs, whereas the direct AMPs develop into the DFMs. Control adults displayed wild-type posture (*Figure 5I*), while after Hh signaling was downregulated in AMPs with *smo$^{RNAi}$*, we found that a majority of adults displayed an 'outstretched' wing posture phenotype (*Figure 5J, L*). When *ci$^{3m}$* was expressed in all AMPs, we observed that many adults displayed a 'downtilted' wing posture (*Figure 5K, L*). These wing posture phenotypes were reproducible with multiple *smo$^{RNAi}$* and in both sexes (*Figure 5—figure supplement 2A*). Adults with either the outstretched or downtilted phenotypes were incapable of flight. These observations suggest a crucial role of Hh signaling within the AMPs for the formation of functional adult flight muscles.

To examine if Hh-signaling perturbations affected the structure of adult muscle fibers, we dissected adult thoraxes (*Figure 5M–R*). When we reduced *smo* expression in the AMPs, we observed a misalignment of the DFM fibers (*Figure 5U*). In particular, the more posterior DFMs 52–57 (*Miller, 1950*; *Bate, 1993*; *Ghazi et al., 2000*) displayed improper position and overall disorganization (*Figure 5—figure supplement 2B–M*). Muscle 53, for example, inappropriately projects to the dorsal attachment site of muscle 54. In contrast, the DLM and DVM muscle fibers appeared relatively normal (*Figure 5S, T*). This indicates that the loss of Hh signaling within the AMPs causes defects in the adult muscles, specifically a subset of muscles formed by the direct AMPs.

Conversely, when the Hh-signaling pathway was constitutively active via the expression of *ci$^{3m}$* within the AMPs, we observed elimination of the DVMs (*Figure 5W*) and the DFMs were often severely disorganized and malformed (*Figure 5X*). Importantly, muscle 51, which is derived from a

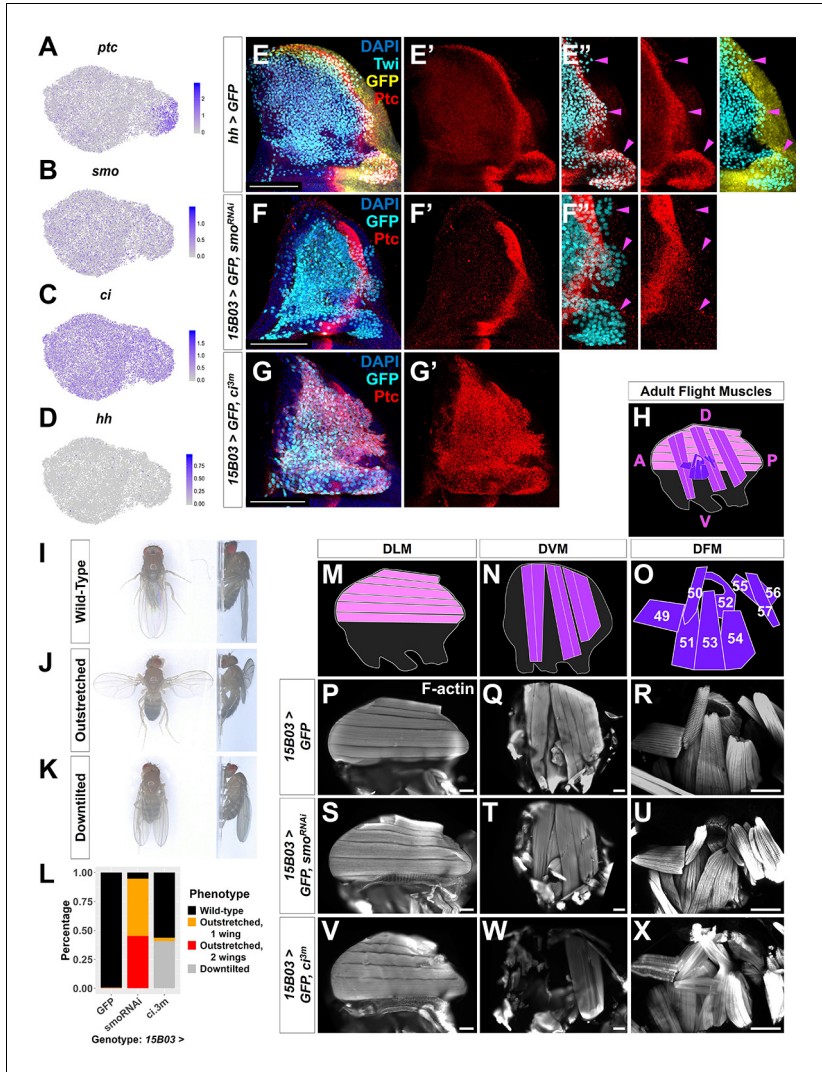

**Figure 5.** Hh signal from the disc epithelium patterns a subset of adult muscle precursors (AMPs). (A–D) UMAPs of *ptc* (A), *smo* (B), *ci* (C), and *hh* (D) expression in AMPs. (E–E'') Notum of wing discs stained for anti-Ptc (red) with the *hh*-expressing epithelium cells marked by *hh-Gal4* driving *>GFP* (yellow). AMPs visualized by anti-Twi stain (cyan). Note Ptc expression in the posterior-localized AMPs, to the right of the epithelium Ptc stripe and in close proximity to the *hh*-expressing epithelium (magenta arrowheads) (E'') (see *Figure 5—figure supplement 1*). (F–G') Notum of wing discs stained for anti-Ptc (red) with *15B03-Gal4* driving *>GFP* together with either *>smo^RNAi* (BL43134) to reduce Hh signaling within the AMPs (F–F'') or *>ci^3m* to mimic activated Hh signaling within the AMPs (G, G'). Ptc expression is significantly reduced in the posterior-localized AMPs after *smo* knockdown (magenta arrowheads) without disrupting the stripe of Ptc expression in the epithelium (F''). Note that Ptc is expressed in all of the AMPs following *>ci^3m* expression (G'). (H) Schematic of adult flight muscle fibers within the thorax where the muscle subtypes are differentially shaded: dorsal longitudinal muscles (DLMs) in pink, dorsoventral muscles (DVMs) in purple, and direct flight muscles (DFMs) in dark purple. (I–K) Wing posture phenotypes observed following Hh-signaling perturbations. Adults were imaged live, not anesthetized. (I) Wild-type posture, with wing blades folded along their dorsum. (J) Outstretched wing posture, where either one or both wings were always held perpendicular to the body axis. (K) Downtilted wing posture, with adults that hold their wings farther apart along their dorsum and tilted laterally downward. (L) Quantification of wing posture after Hh-signaling perturbation within AMPs (driven by *15B03-Gal4*). The number of adults assayed: *>GFP*=341, *>smo^RNAi* (BL43134) = 283, and *>ci^3m* = 366. These *smo^RNAi* results were replicable with multiple RNAi lines (*Figure 5—figure supplement 2A*). (M–O) Separate schematics of expected DLMs (M), DVMs (N), and DFMs (O) morphology. Numbers on DFMs represent the canonical labels for the different fibers. (P–X) Adult flight muscles (visualized with F-actin staining) from animals with *15B03-Gal4* driving *>GFP* alone (P–R), or *>GFP* together with either *>smo^RNAi* (S–U) or *>ci^3m* (V–X). DLMs are shown in (P), (S), and (V); DVMs are shown in (Q), (T), and (W); DFMs are shown in (R), (U), and (X).
*Figure 5 continued on next page*

*Figure 5 continued*

Adult flight muscles in >*GFP* flies had similar morphology in all adults examined (23 DLMs, 15 DVMs, and 12 DFMs). Adult flight muscles in >*smo*$^{RNAi}$ (BL43134) animals displayed abnormal DFMs (11/11 had muscles 53 and 54 misaligned, and 7/11 had muscles 55, 56, and 57 malformed) (**U**), while DLMs (n = 21/22) and DVMs (n = 10/10) had relatively normal morphology (**S, T**). Adult flight muscles in >*ci*$^{3m}$ animals had normal DLMs morphology (n = 7) (**V**), whereas the DVMs were either missing or severely disconnected (n = 7) (**W**) and the DFMs appeared abnormal (n = 4) (**X**). UMAP color scales correspond to normalized counts on a natural-log scale. All notum images are max projections across image slices containing AMPs. Microscopy scale bars = 100 μm.

The online version of this article includes the following figure supplement(s) for figure 5:

**Figure supplement 1.** Ptc-expressing adult muscle precursors (AMPs) are neighboring the Hh-producing posterior compartment of the disc proper.

**Figure supplement 2.** Adult wing-posture phenotypes and morphology of individual muscle fibers after Hh-signaling perturbation.

---

separate group of AMPs not associated with the wing disc (*Lawrence, 1982*), is unaffected by these manipulations. The DLMs had no noticeable defects (*Figure 5V*), likely because the DLMs do not arise in de novo, unlike the other adult flight muscles, but rather by the fusion of AMPs with histolyzing larval muscles that act as templates (*Fernandes et al., 1991*). Overall, our data suggest that Hh signaling is important for proper specification of a subset of the direct AMPs and that excessive Ci activity causes inappropriate patterning that perturbs the development of the IFMs.

## Neurotactin and midline are AMP-specific downstream targets of Hh signaling

What are the downstream targets that are activated by Hh signaling in the AMPs? Since the canonical target of Hh signaling in the disc epithelium, *dpp*, is not expressed in the AMPs, we searched for candidate genes that were specifically expressed within the subpopulation of AMPs that express high levels of *ptc*. We found that *midline* (*mid*) and *Neurotactin* (*Nrt*) displayed relatively high correlation with that of *ptc* (Pearson correlation of 0.44 and 0.34 with *mid* and *Nrt*, respectively) (*Figure 6A–C*). Mid, also known as Neuromancer 2, is a T-box transcription factor most related to mouse Tbx-20 (*Buescher et al., 2004*) that regulates cell fate in the developing nervous system (*Leal et al., 2009*). Nrt encodes a single-pass transmembrane protein expressed on the cell surface (*Hortsch et al., 1990*). Dimers of the secreted protein Amalgam (Ama), which are expressed in the direct AMPs, are able to bind to Nrt on two different cells and promote their adhesion (*Frémion et al., 2000*; *Zeev-Ben-Mordehai et al., 2009*).

*Nrt* and *mid* expression increased dramatically in direct AMPs from 96 to 120 hr based on our single-cell data (*Figure 6A, B*) and antibody staining (*Figure 6D–G*), whereas *ptc* is expressed at similar levels at both time points (*Figure 6C, H, I*). Surprisingly, while Nrt and Mid expression patterns included all posterior-localized direct AMPs, the expression of both genes extended into anterior-localized direct AMPs, which do not express Ptc. Due to their high expression levels in the posterior-localized AMPs, we hypothesized that expression of Nrt and Mid, at least in the posterior AMPs, was influenced by Hh signaling.

To determine if *Nrt* and *mid* are downstream Hh-signaling targets, we examined if perturbing the Hh pathway within the AMPs would alter their expression. The knockdown of *smo* and consequent reduction in Hh signaling resulted in a dramatic decrease in both Nrt and Mid expression in the direct AMPs at 120 hr AEL (*Figure 6J, K, N*). Remarkably, this was observed in both the posterior- and anterior-localized AMPs alike. We tested if increased Hh signaling would be sufficient to induce expression of Nrt and Mid by driving *ci*$^{3m}$ in the AMPs. This resulted in the ectopic expression of both Nrt and Mid in all of the AMPs, although expression was higher within direct AMPs (*Figure 6L–N*). Thus, all AMPs are capable of inducing Nrt and Mid expression in response to Hh signaling, but only the AMPs closest to Hh-secreting epithelial cells do so under physiological conditions. This relationship between *ptc*, *mid*, and *Nrt* is specific to the AMPs as we did not observe correlation between *ptc* and either *Nrt* or *mid* within the epithelium. These experiments indicate that Hh signaling is important for proper patterning of the AMPs. However, Hh signaling is not required to regulate all aspects of the direct AMP cell fate as high Ct expression was unaffected following the manipulation of Hh signaling by *smo* knockdown (*Figure 6—figure supplement 1A, B*). This

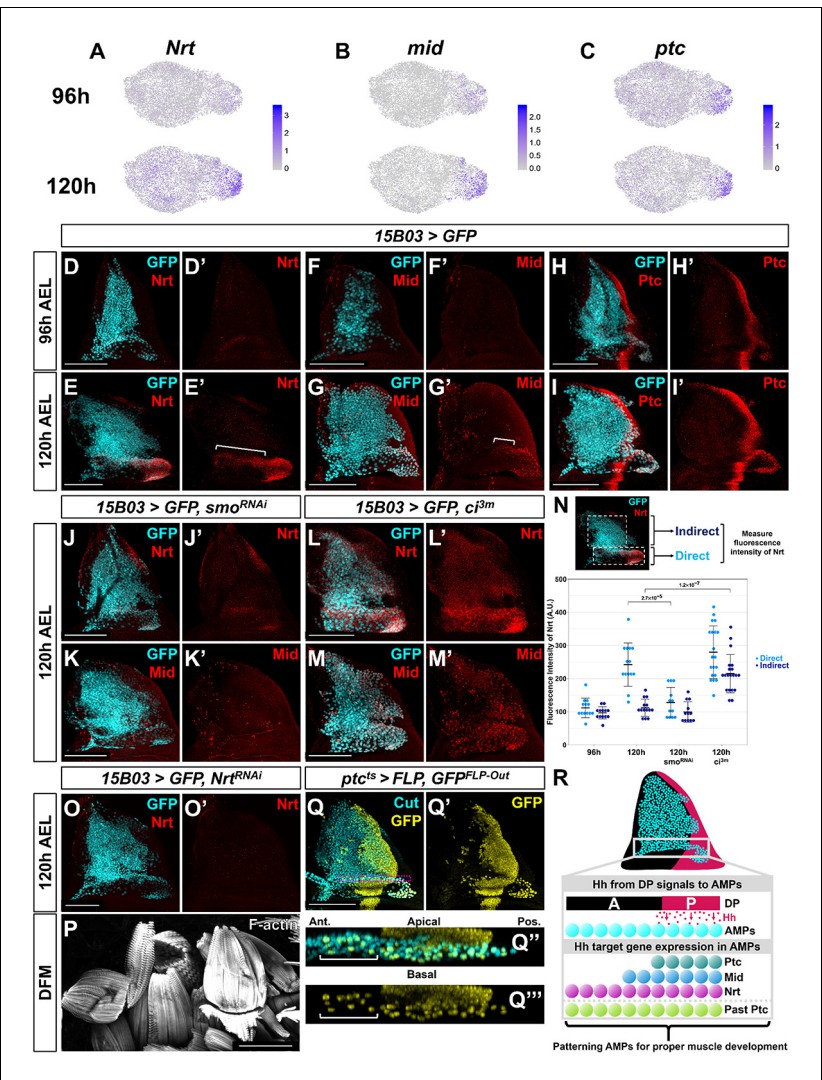

**Figure 6.** Nrt and Mid are novel downstream Hh-pathway targets in the adult muscle precursors (AMPs). (**A–C**) UMAPs of *Nrt* (**A**), *mid* (**B**), and *ptc* (**C**) expression at 96 and 120 hr in AMPs. Note the increase in expression for *Nrt* and *mid* from 96 to 120 hr, whereas *ptc* expression is relatively unchanged. (**D–I'**) Wing discs stained with anti-Nrt at 96 hr (**D, D'**) and 120 hr (**E, E'**), anti-Mid at 96 hr (**F, F'**) and 120 hr (**G, G'**), and anti-Ptc at 96 hr (**H, H'**) and at 120 hr (**I, I'**). AMPs are visualized by expression of GFP (cyan) via *15B03-Gal4* driver. Note the negligible staining of anti-Nrt and anti-Mid in AMPs at 96 hr, matching the scRNAseq expression data. (**J–M'**) Wing discs expressing either >*smo*<sup>RNAi</sup> to reduce Hh signaling within the AMPs (**J–K'**) or >*ci*<sup>3m</sup> to activate Hh signaling within the AMPs (**L–M'**) via *15B03-Gal4* driver, along with >*GFP* to visualize AMPs (cyan). Discs are either stained with anti-Nrt (**J, J', L, L'**) or anti-Mid (**K, K', M, M'**). Note that the knockdown of *smo* prevents the expression of Nrt and Mid in the direct AMPs and that the overexpression of *ci*<sup>3m</sup> leads to ectopic expression in the indirect AMPs. (**N**) Quantification of anti-Nrt staining within direct and indirect AMPs. The graph shows binned values of average fluorescent intensity. p-Values were calculated from unpaired *t*-tests, and error bars indicate standard deviation. Note that *smo* knockdown prevents Nrt expression within the direct AMPs at 120 hr, and the expression of activated *ci* increases Nrt expression in both the direct and indirect AMPs. (**O, O'**) Wing disc expressing >*Nrt*<sup>RNAi</sup> in AMPs via *15B03-Gal4* driver, stained with anti-Nrt at 120 hr. Note that the knockdown of *Nrt* eliminates Nrt staining in the AMPs (**O'**). (**P**) Direct flight muscles (DFMs) in adults where *15B03-Gal4* drives >*Nrt*<sup>RNAi</sup>. Note the enlarged posterior DFMs, specifically muscles 55, 56, and 57 (similar phenotypes were observed in all five flies examined) (compare to the control in *Figure 5R*). (**Q–Q'''**) Lineage tracing of AMPs that have previously expressed *ptc* earlier in development; genotype: *ptc-Gal4, tub-Gal80*<sup>ts</sup>, >*FLP, Ubi-FRT-stop-FRT-GFP* (*ptc*<sup>ts</sup>>*FLP, GFP*<sup>FLP-Out</sup>). Cells that expressed *ptc-Gal4* when larva are shifted to higher temperature (non-permissive temperature for Gal80<sup>ts</sup>) and their progeny will be permanently labeled by GFP expression. The temperature shift from 18°C to 30°C was done at 5 days after egg lay for 24 hr, larvae were dissected at late third instar. AMPs were visualized with

*Figure 6 continued on next page*

*Figure 6 continued*

anti-Cut staining. Orthogonal max projection is shown in (**Q''**) and (**Q'''**), corresponding to the dashed purple box in (**Q**). Note that a subset of the anterior labeled AMPs (indicated by white bracket) expresses GFP. (**R**) Model explaining the protein expression of Hh-signaling targets in AMPs. Posterior-localized AMPs receive Hh from the disc proper, activating expression of Hh-signaling targets Ptc, Nrt, and Mid (the latter two being AMP-specific targets). These AMPs migrate anteriorly, either a result of active cell movement or due to displacement caused by proliferation. Anterior-localized AMPs quickly degrade Ptc protein, but Nrt and Mid perdure longer. Microscopy scale bars = 100 μm.

The online version of this article includes the following figure supplement(s) for figure 6:

**Figure supplement 1.** Knockdown of *smo* does not affect Ct protein levels in adult muscle precursors (AMPs).

suggests that Hh signaling is acting independent from *ct* to specify the fate of a subpopulation of the AMPs.

To evaluate the functional consequences of reducing *Nrt* and *mid* expression, we used RNAi to reduce their expression. The RNAi line directed against *mid*, however, failed to significantly reduce Mid protein levels. In contrast, *Nrt* knockdown reduced Nrt levels (**Figure 6O**) and resulted in adults with defects in the posterior DFMs (**Figure 6P**; compare to **Figure 5R**), albeit not as severely as knockdown of *smo*. This result shows that the Hh-signaling downstream target *Nrt* is critical for proper DFM development.

The AMPs with high Ptc expression are in the vicinity of Hh-expressing epithelial cells, which is consistent with the notion that Hh is a short-range morphogen capable of acting on cells within 40 μm from the source. However, we observed expression of the two target genes, Nrt and Mid, in anterior-localized AMPs far beyond 40 μm from the Hh source. Knockdown of Hh signaling via *smo*[R-NAi] expression abolished the expression of both genes in anterior- and posterior-localized AMPs alike (**Figure 6J, K**), ruling out the possibility of their activation in a Hh-independent manner. This raised the possibility that these cells were initially closer to the Hh source earlier in development and had subsequently moved anteriorly either as a result of passive displacement or active migration. In this scenario, the expression of Nrt and Mid would be expected to perdure for longer than the expression of Ptc. To test whether anterior-localized AMPs had past activation of the Hh-signaling pathway, we used a lineage-tracing method to identify cells that had previously expressed *ptc* (*ptc*[ts]>*FLP, GFP*[FLP-Out]; see Materials and methods). We labeled all cells that displayed activity of *ptc-Gal4* during late second and early third instar, and observed lineage labeling (indicated by GFP expression) of all direct AMPs underlying the posterior compartment of the epithelium as well as a trail of anterior AMPs that recapitulate the domain of anti-Nrt staining (**Figure 6Q**). These results indicate that these anterior GFP-positive AMPs likely descended from cells that previously expressed high levels of *ptc*, and that past activation of the Hh pathway is likely to be responsible for the expression of both anterior- and posterior-localized Nrt and Mid (**Figure 6R**).

## Discussion

### Heterogeneity and diversification of cell types

Data obtained from scRNAseq experiments can provide both spatial and temporal information that give us a better understanding of how cells diversify and then stabilize their transcriptomes during development, and point to ways in which they interact with each other. Moreover, we have been able to visualize the expression of genes from our dataset in a three-layered virtual wing disc – a format that will be useful for developmental biologists.

One interesting observation from our data is that spatial positioning within the wing disc is highly informative of the transcriptional state of cells. In particular, the proximodistal axis of the disc epithelium is one of the primary stratifying features within our single-cell data. Epithelial cell clusters were easily mapped back to sub-regions within the notum, hinge, pouch, and PE. In contrast, although the cells of the anterior and posterior compartments have been separated by lineage since early in embryogenesis, we observe less differential expression between the two compartments. Thus, position along the proximodistal axis has a far greater influence on the transcriptome of a cell than its ancestry.

In the epithelium, we observe that most of the major cell types observed at late L3 (120 hr AEL) are already present at mid L3 (96 hr AEL). However, the transcriptomes of the two major populations of AMPs, those that give rise to the DFMs and IFMs, diverge significantly during this time interval. At 96 hr, AMPs appear to be in a relatively naive state; canonical markers for the direct and indirect flight precursors, *ct* and *vg*, both show relatively uniform expression at the mid L3 stage. At the late L3 stage, we observe more distinguishable differences between the transcriptomes of direct and indirect cell types. Both *ct* and *vg* have greater differential expression in the AMPs at this time point. The earlier stabilization of epithelial cell fates then provides a stable platform for the generation of spatially localized signals that regulate myoblast numbers and provide instructive signals for fate specification.

## FGF signaling regulates the number and location of AMPs

Ths and Pyr have previously shown to regulate the spreading of mesodermal cells during embryogenesis, but a role for these ligands in regulating myoblast numbers was not previously appreciated. We show these ligands are necessary for AMP survival and that increased levels of these ligands can promote AMP proliferation even at sites distant from the notum. Indeed, increased expression can induce a dramatic overproliferation of the AMPs, even as late as third instar. Ectopic FGF signaling is sufficient to support AMP viability and cell proliferation at other locations in the wing disc, including underneath the pouch. Thus, the localized expression and level of Ths and Pyr secreted by epithelial cells in the notum could provide sufficient trophic support to generate the appropriate number of AMPs during normal development. While this work was in preparation, another group independently showed that the *ths-Gal4* line is expressed in the notum epithelium and that reducing *ths* function reduces AMP numbers (*Vishal et al., 2020*).

We have also demonstrated that ectopic and elevated levels of expression of Ths or Pyr can draw AMPs out of the notum region, all the way to the ventral hinge and around the ventral edge of the disc proper onto the peripodial epithelium. With constitutive FGF ligand expression driven by *dpp-Gal4*, we did not observe AMP migration to the lateral regions of the disc, which are far from the ectopic FGF source. However, when FGF expression is initiated in L3 by controlled Gal80[ts] repression, we observed a large number of laterally located AMPs. We attribute the difference in these two scenarios to ectopic AMPs along the *dpp* stripe possibly serving as an FGF sink. When FGF is turned on later in development, there are no ectopic AMPs. In their absence, FGF could reach more lateral portions of the disc, thus allowing AMP emigration to those regions. Furthermore, the expression of Pyr in the wing pouch, which is separated from the notum by the dorsal hinge, was sufficient to promote colonization of the pouch region by AMPs. The AMPs can be induced to colonize new domains late into larval development by the expression of either FGF ligand. These results suggest that under physiological conditions AMPs remain beneath the notum epithelium because there is insufficient FGF outside of this region. Altogether, our work illustrates that a source of epithelial FGF is critical for forming the AMP niche and that levels of FGF regulate both the location and number of the AMP cells.

## Instructive Hh signaling from the epithelium to the myoblasts

We have shown that the anteroposterior identities of the disc epithelium are important for proper specification of gene expression within the underlying AMPs. One powerful advantage of scRNAseq as opposed to its bulk sample predecessor is the ability to measure gene co-expression within subpopulations of a tissue. We leveraged this advantage to identify two novel Hh-signaling targets, *Nrt* and *mid*, within AMPs. While we currently do not know whether *Nrt* and *mid* are direct targets of Ci, both genes do have consensus Ci-binding sites within potential regulatory regions. Nrt is a single-pass transmembrane protein. Its extracellular ligand, Amalgam, has more widespread expression in the direct myoblasts and is expressed at comparable levels at both 96 and 120 hr AEL. Two molecules of Amalgam can form homodimers, and each is capable of binding to Nrt on different cells (*Frémion et al., 2000*; *Zeev-Ben-Mordehai et al., 2009*). Thus, an effect of Hh-induced expression of Nrt in a subset of the direct adult flight muscles might be to promote aggregation of Nrt-expressing cells at a later stage of development.

An unexpected observation was that AMPs beneath the anterior compartment, distant from the epithelial source of Hh, express both identified Hh targets Nrt and Mid. However, this expression is

dependent upon Hh signaling since knockdown of *smo* blocks gene expression. Although we cannot completely exclude the possibility that a second signal from posterior AMPs activates *Nrt* and *mid* expression in these cells, our lineage-tracing experiments favor a model where a subset of the direct AMPs are generated posteriorly and move anteriorly during the course of development. Such movement could be due to a process of active migration in response to hitherto unknown external cues or to displacement as a result of oriented cell division. Understanding the mechanistic basis of AMP migration would represent an exciting avenue of future research.

## Concluding remarks

Our work has provided a base for the study of heterotypic interactions in the developing wing disc during conditions of normal growth and demonstrate that such interactions can have a major effect on cell number, cell migration, and cell fate in the wing disc. By examining receptor–ligand expression patterns in conjunction with spatial mapping of our single-cell data, our analysis provides many hints of signaling pathways that may function between the disc epithelium and the AMPs and also within subsets of cells with each of these populations that provide multiple avenues for future investigations.

# Materials and methods

## Generation of single-cell suspension, barcoding, and sequencing

For each sample, approximately 250 staged *Drosophila* wing-imaginal discs were dissected within 1 hr. The collected tissue was then transferred to a microcentrifuge tube and incubated within a dissociation cocktail consisting of 2.5 mg/mL collagenase (Sigma #C9891) and 1× TrypLE (Thermo Fisher #A1217701) in Rinaldini solution (modified from *Ariss et al., 2018*). The sample tube was placed horizontally on a shaker machine operating at 225 rpm for 25 min at room temperature (method modified from *Ariss et al., 2018*). At the 10, 20, and 25 min marks, the tube was flicked 20 times for additional mechanical dissociation. Dissociation was halted by centrifuging the sample at 5000 rpm for 3 min, aspirating the dissociation cocktail, and then adding in 1 mL of cold PBS-10% FBS. The cell pellet was mixed by pipetting up-and-down approximately 25 times with a 1 mL pipette for additional mechanical dissociation, and then centrifuged again at 5000 rpm for 3 min. The media was replaced with cold PBS-1% FBS, and the cell pellet was resuspended in preparation for FACS.

FACS of the sample was performed on a BD FACSAria Fusion flow cytometer. Dead cells were identified and removed via the addition of propidium iodide to the sample, and high-quality single cells were sorted into cold PBS-10% FBS. Cell concentration of the post-FACS sample was assessed by a hemocytometer and adjusted 1000 cells per μL.

Single-cell suspensions were barcoded for single-cell RNA sequencing with the 10X Chromium Single Cell platform (v2 chemistry). Barcoded samples were sequenced on an Illumina NovaSeq (S2 flow cell) to over 60% saturation.

## Single-cell data processing and analysis

The 10X Genomics Cell Ranger pipeline (v2.2.0) was used to align the sequencing reads to the *Drosophila melanogaster* transcriptome (version 6.24). The data was analyzed using the R and Python programming languages, primarily utilizing the packages Seurat v3 (*Stuart et al., 2019*) and scVI v0.4.1 (*Lopez et al., 2018*).

Our standard analysis pipeline is as follows: first, each dataset was analyzed separated using the standard Seurat pipeline, with no cells filtered, 30 principal components calculated, and clustering resolution set to 2.0 (all other parameters remained default). We then removed cell clusters with an abundance of low-quality cells (defined as clusters with mean number of genes detected per cell [nGene] was less than one standard deviation below the mean nGene of all cells in the dataset). Additionally, we found that each dataset had a cluster with markers for both AMP and epithelial cell types (e.g., *SPARC* and *Fas3*) and unusually high mean nGene; this cluster was suspected to be AMP-epithelial doublets and was also removed. Clusters were then split into AMP and epithelial cell subsets based on the expression of known marker genes. Cells within each subset were subsequently filtered if either (1) their nGene count that was outside the mean nGene of the subset ±1.5

standard deviations or (2) their percentage of reads for mitochondrial genes that was greater than 1.5 standard deviations above the mean mitochondrial read percentage of the subset.

Data subsets were harmonized into collective AMP or epithelium datasets using scVI. The scVI VAE model consisted of 2 layers (n_layers = 2) and 20 latent dimensions (n_latent = 20), with a negative-binomial reconstruction loss (reconstruction_loss='nb'). The model was trained on variable genes selected by Seurat's variance-stabilizing transformation method; 1000 (for epithelial subsets) or 2000 (for AMP subsets) variable genes were calculated for each inputted batch, and then the union of these genes was supplied to scVI. The following parameters were used for model training: train_size = 0.75, n_epochs = 400, and lr = 1e-3 (other parameters were left as default). Cell clustering and UMAP was performed using Seurat on the latent space derived from the scVI model. After harmonization, clusters were re-examined for doublet characteristics; clusters with a mean nGene count greater than one standard deviation above the mean nGene count of all cells were removed, as were clusters that displayed markers for both AMP and epithelial cell types. Identified hemocyte and tracheal cells were also separated out. scVI and Seurat were both re-run on the datasets to generate our final AMP and epithelial cell atlases (*Figure 1H, N*).

To generate our full cell atlas consisting of all cell types (*Figure 1B, C*), we merged and harmonized the cells in the AMP and epithelium cells atlases along with the separated hemocyte and tracheal cells. scVI and Seurat were run as previously described, with the scVI model trained on the union of the top 2000 variable genes for each batch as calculated by Seurat. No additional cell filtering was performed after harmonization.

For visualizing data on UMAPs and dot plots, we calculated normalized and scaled expression counts using Seurat's NormalizeData and ScaleData functions, respectively, with default parameters. For the normalized data, raw counts were normalized by total unique molecular identifiers (UMIs) per cell, multiplied by 10,000. Natural-log normalized data is used for expression levels visualized with UMAP. For the scaled data, the natural-log normalized data is scaled for each gene, such that the mean expression is 0 with a standard deviation of 1. Scaled data is used for expression visualization on the dot plots.

## Cell sex and cell cycle correction with AMP data

Cells were classified as male or female by their expression levels of the dosage compensation complex genes *lncRNA:roX1* and *lncRNA:roX2* (*Franke and Baker, 1999*; *Meller and Rattner, 2002*), which are both expressed almost exclusively in male cells. For both genes, we examined the natural-log normalized expression counts (calculated by Seurat's NormalizeData function), computed the density over the data, and identified the first local minima as a threshold (see *Figure 1—figure supplement 4C, D*). Cells that were above the threshold for either *lncRNA:roX1* or *lncRNA:roX2* were classified as male; otherwise, they were classified as female. From this, we assigned 8097 cells as male and 11,788 cells as female, which roughly matches the size ratio between male and female wing discs given that female discs are larger. We removed cell sex stratification by processing male and female AMPs as separate batches (for each actual batch) within scVI (see *Figure 1—figure supplement 4E, F* for comparison of data before and after cell sex correction).

We observed significant data stratification that correlated with a number of cell cycle-related genes, such as *proliferating cell nuclear antigen* (*PCNA*) and *cyclin B* (*CycB*) (*Yamaguchi et al., 1990*; *Lehner and O'Farrell, 1990*), indicating that our data was split between S phase and non-S phase (*Figure 1—figure supplement 5A–C*). Definitive classification of cells into cell cycle stages is difficult because expression of these genes is not typically demarcated sharply into specific cell cycle stages. To remove cell cycle stratification from our data, we examined the correlation of each scVI latent dimension with the expression levels of highly variable cell cycle genes and found that one latent dimension was strongly related (*Figure 1—figure supplement 5D*). By masking this latent dimension from our downstream analysis (e.g., clustering and UMAP), we effectively diminished cell cycle stratification. *Figure 1N* shows the UMAP of our AMP data after subtraction of cell sex and cell cycle stratification, which allowed us to focus our analysis on different cell types within the AMPs (to see how each correction affected the AMP data, see *Figure 1—figure supplement 5A–C*).

## Determining differentially expressed genes

When examining clusters, genes were considered to have significant differential expression if they had (1) a false discovery rate (FDR) <0.05 (as calculated via Wilcoxon test), (2) a natural-log fold-change of 0.15 or more, and (3) a percent expression of at least 15% in one of the two populations in the comparison. This test was performed with Seurat's FindMarkers function, using a Wilcoxon test (test.use = 'wilcox').

When evaluating differential expression between clusters of epithelial cells (e.g., *Figure 1—figure supplement 2A*), we used a one cluster vs. all analysis. When evaluating differential expression between direct and indirect AMPs (e.g., *Figure 1—figure supplement 5E*), we compared cells of the two groups as classified in *Figure 1—figure supplement 5F*. In these cases, differential expression statistics (i.e., FDR and fold-change) are obtained by combining the cells (across batches) in each group and conducting a single comparison. When evaluating differential expression between time points (e.g., *Figure 1—figure supplement 1H*, *Figure 1—figure supplement 2C*, *Figure 1—figure supplement 6A*) (which would be inherently confounded with batch effects since time points were collected across separate sequencing experiments), we took a conservative approach and only considered genes that were consistently significant (by the criteria defined above) in each temporal pairwise comparison (i.e., DE analysis was conducted between all temporal pairs: 96h1 vs. 120h1, 96h2 vs. 120h1, 96h2 vs.120h1, 96h2 vs.120h2). We report the natural-log of the average value for these pairwise comparisons, and the maximum FDR calculated (see *Supplementary files 1–3*).

## Generating a virtual model of the wing disc

We assembled reference gene expression patterns from a number of sources (*Held, 2002*; *Butler et al., 2003*) and based our starting geometry on the disc proper from images in *Bageritz et al., 2019*. The images were processed in Adobe Photoshop and assembled in R with EBimage (*Pau et al., 2010*) to generate binarized gene expression reference for the AMPs, disc proper, and peripodial epithelium. The geometry of the three-layered model is provided in *Supplementary file 4,* and the binarized reference gene expression patterns are provided in *Supplementary file 5*. We used DistMap (*Karaiskos et al., 2017*) to statistically map single cells back to the reference. With this virtual wing disc model, we used DistMap to calculate a 'virtual in situ' or a prediction of gene expression patterns. This is based on the detected gene expression with the single-cell data and the mapping location to calculate relative expression values for our model. We mapped the AMP and epithelial cells separately as this improved how well the model predicted genes with known expression patterns. In addition, we used the scVI imputed gene expression values when mapping the cells to the reference.

## Examination of receptor–ligand expression

From FlyBase, we assembled a list of genes encoding receptors and ligands from the following 19 pathways of interest: Wnt/Wingless, FGF, Hedgehog, PDGF/VEGF, JAK-STAT, activin, BMP, Fat-Ds, Slit-Robo, ephrin, toll/toll-like, semaphorin, Notch, insulin-like, fog, torso, miple, EGFR, and TNF. For our analysis, we only examined pathways in which at least one receptor or ligand was either (1) differentially expressed within one of the major domains of the epithelium (notum, hinge, pouch, or PE) when compared to all other epithelial cells, (2) differentially expressed within one of the major domains of the AMPs (direct or indirect cells) when compared to each other, or (3) differentially expressed between all epithelial cells vs. all AMP cells. These pathways (and their receptors and ligands) are shown in *Figure 4H*.

## *Drosophila* stocks and husbandry

The stocks used in this study include the following lines from the Bloomington Stock Center: *R15B03-GAL4* (BL49261); *G-TRACE* (BL28280, 28281) (*Evans et al., 2009*); *UAS-FLP, Ubi-FRT-stop-FRT-GFP$^{nls}$* (BL28282); *pdm3-GFP* (BL60560); *grn-GFP* (BL58483); *ptc-GAL4* (BL2017); *tub-GAL80$^{ts}$* (BL7108); *dpp-GAL4* (BL1553); *dpp-GAL4, tub-GAL80$^{ts}$, UAS-dCas9.VPR* (BL67066); *nub-GAL4* (BL25754); *hh-GAL4* (*Tanimoto et al., 2000*); *ap-GAL4* (BL3041); *fng-Gal4* (BL9891); *htl-GAL4* (*GMR93H07-GAL4,* BL40669) is an enhancer within the first intron of the *htl* gene; *UAS-smo$^{RNAi}$* (primarily BL43134, but also BL27037, 62987 in *Figure 5—figure supplement 2*); *UAS-Nrt$^{RNAi}$* (BL28742). *Drosophila* stocks from other labs: *UAS-ths* and *UAS-pyr* (A Stathopoulos); *UAS-ci$^{3m}$* (D

Kalderon). TRiP-CRISPR driven overexpression of *pyr* was conducted with a guide RNA that targets the upstream transcriptional start site, *P{TOE.GS00085}attP40* (BL67537), and works together with a nuclease-dead Cas9 fused with a transcriptional activator domain, *UAS-dCas9.VPR* to cause gene activation (BL67055) (*Lin et al., 2015*).

## Dissections, immunohistochemistry, and microscopy

Imaginal discs, unless otherwise noted, were fixed in 4% paraformaldehyde (PFA) for 15 min, permeabilized in PBS plus 0.1% Triton X-100, and blocked in 10% normal goat serum. For anti-Nrt antibody staining, we substituted the Triton X-100 for 0.05% saponin. The following antibodies were used from the Developmental Studies Hybridoma Bank (DSHB): mouse anti-Cut (1:200, 2B10); mouse anti-Ptc (1:50, Apa-1); mouse anti-Nrt (BP 106 anti-Neurotactin); and mouse anti-Wg (1:100, 4D4). The following antibodies were gifted: rat anti-Twist (1:1000, Eric Wieschaus), rabbit anti-Midline (1:500, James Skeath), and rat anti-Zfh2 (1:100, Chris Doe; *Tran et al., 2010*). The following antibodies are from commercial sources: rabbit anti-Dcp1 (1:250, Cell Signaling); rabbit anti-GFP (1:500, Torrey Pines Laboratories, Secaucus, NJ); chicken anti-GFP (1:500, ab13970 Abcam, Cambridge, UK); rabbit anti-beta-galactosidase (1:1000, #559762; MP Biomedicals, Santa Ana, CA); and rabbit anti-PHH3 (1:500, Millipore-Sigma). Secondary antibodies were from Cell Signaling. Nuclear staining with DAPI (1:1000).

For EdU staining, we incubated live discs in fluorescent EdU incorporation solution for 1 hr, following the protocol for the Click-iT EdU Cell Proliferation Kit, Alexa Fluor 555 (ThermoFisher C10338). After the incubation, discs were fixed in 4% PFA for 15 min, before proceeding with standard antibody stainings as detailed above.

To ectopically express FGF ligands starting at mid or late third instar development (e.g., *Figure 4G, H*), we used a temperature-sensitive *dpp-Gal4* stock (*dpp-GAL4, tub-GAL80$^{ts}$, UAS-dCas9.VPR*) (BL67066) with *UAS-pyr* and *UAS-ths* lines. In-vial egg lays were collected over 8 hr, and larvae were initially raised at 18°C. Larvae were shifted to 30°C (to relieve Gal80$^{ts}$ repression) at either 48 or 24 hr prior to dissection, corresponding to mid and late third instar FGF activation, respectively.

To lineage-trace cells that expressed *ptc-Gal4* during the second and early third instars, we used a temperature-sensitive, Ptc-dependent FLP-Out system (*ptc-Gal4, tub-Gal80$^{ts}$, UAS-FLP, Ubi-FRT-stop-FRT-GFP$^{nls}$*). Under this system, inactivation of *Gal80$^{ts}$* repression permits cells that express *ptc-Gal4* (and their descendant cells) to become permanently labeled with GFP by an FLP–FRT recombination event. In the experiment shown in *Figure 6Q*, first instar larvae were collected and reared at 18°C. At 5 days post-egg lay, larvae were shifted to 30°C for 24 hr, then shifted back to 18°C for 48 hr prior to dissection.

Wing discs were imaged on a Zeiss Axioplan microscope with Apotome attachment, using ×10 and ×20 objectives. Image files were processed with ImageJ software. For each of the genotypes examined, we examined at least eight discs and have reported representative results in this paper.

## Adult muscle preparations

To image adult flight muscles, male flies aged a minimum of 2 days after eclosion were anesthetized and submerged in 70% ethanol with dry ice. The thorax was isolated by removing the head, wings, legs, and abdomen. Thoraces were bisected sagittally with a 11-blade scalpel blade. For DVMs, the DLMs, leg muscles, and excess cuticle were removed from hemithoraces. For DFMs, the DLMs, DVMs, leg muscles, and excess cuticle were removed from hemithoraces. The DLMs, DVMs, and DFMs were fixed in 4% PFA for 2 hr. Muscles were then rinsed three times and permeabilized in 0.3% PBST for three cycles, 15 min each on a nutator. Hemithoraces were incubated in rhodamine phalloidin (1:200) and DAPI (1:500) in 0.3% PBST, then rinsed three times and washed in 0.3% PBST for three cycles, 15 min each on a nutator. Hemithoraces were mounted in a depression slide using antifade mountant. DLMs and DVMs were imaged with a ×10 objective using a Zeiss Axioplan microscope. DFMs were imaged with the ×20 and ×63 objectives using confocal microscopy.

## Acknowledgements

We thank A Stathopoulos, J Skeath, C Doe, and E Wieschaus for stocks and reagents and M Frolov for discussions. We thank current and former members of the Yosef lab for advice and guidance. We

thank T Ashuach and D DeTomaso for discussions regarding computational analyses. We thank current and former members of the Hariharan lab for advice, guidance, and help with collection of samples. We thank C Wingert for technical assistance. We thank the Bloomington Stock Center, DRSC/TRiP Functional Genomics Resources, and Developmental Studies Hybridoma Bank for stocks and reagents. This work was funded by NIH grant R35 GM122490.

## Additional information

### Funding

| Funder | Grant reference number | Author |
| --- | --- | --- |
| National Institutes of Health | R35 GM122490 | Iswar K Hariharan |

The funders had no role in study design, data collection and interpretation, or the decision to submit the work for publication.

### Author contributions

Nicholas J Everetts, Melanie I Worley, Conceptualization, Data curation, Software, Formal analysis, Validation, Investigation, Visualization, Methodology, Writing - original draft, Writing - review and editing; Riku Yasutomi, Formal analysis, Validation, Investigation, Visualization, Methodology, Writing - review and editing; Nir Yosef, Conceptualization, Software, Formal analysis, Supervision, Project administration, Writing - review and editing; Iswar K Hariharan, Conceptualization, Supervision, Funding acquisition, Writing - original draft, Project administration, Writing - review and editing

### Author ORCIDs

Nicholas J Everetts (ID) https://orcid.org/0000-0001-8897-8481
Melanie I Worley (ID) https://orcid.org/0000-0001-9772-4985
Riku Yasutomi (ID) http://orcid.org/0000-0002-2640-0462
Nir Yosef (ID) https://orcid.org/0000-0001-9004-1225
Iswar K Hariharan (ID) https://orcid.org/0000-0001-6505-0744

### Decision letter and Author response

Decision letter https://doi.org/10.7554/eLife.61276.sa1
Author response https://doi.org/10.7554/eLife.61276.sa2

## Additional files

### Supplementary files

• Supplementary file 1. Genes with differential expression between 96 and 120 hr within the epithelium and adult muscle precursors (AMPs). Genes were selected based on being significantly and consistently upregulated or downregulated between the two time points in either the disc epithelium and/or the AMPs. The average gene expression within cells (natural-log scale), fraction of cells expressing a given gene, fold-change between time points (natural-log scale), and false discovery rate (FDR) for differential expression significance are reported. These gene expression, detection, and fold-change calculations are averaged across each of the pairwise comparisons performed, and the max FDR value is shown (see Materials and methods for details on differential expression between time points). Negative fold-change values indicate higher expression at 96 hr and are colored magenta. Positive fold-change values indicate higher expression at 120 hr and are colored green. N.R.: not replicable; calculations in which the fold-change direction differed between pairwise comparisons.

• Supplementary file 2. Genes with differential expression between 96 and 120 hr within the epithelial cell clusters. Genes were selected based on being significantly and consistently upregulated or downregulated between the two time points in at least one epithelial cluster. The natural-log of the fold-change between 96 and 120 hr is reported, averaged across each of the pairwise comparisons performed (see Materials and methods for details on differential expression between time points).

Negative values indicate higher expression at 96 hr and are colored magenta. Positive values indicate higher expression at 120 hr and are colored green. Values that were not significant (based on max false discovery rate) are reported with a '-'.

• Supplementary file 3. Genes with differential expression between 96 and 120 hr within the direct and indirect adult muscle precursors (AMPs). Genes were selected based on being significantly and consistently upregulated or downregulated between the two time points in either the direct and/or the indirect AMPs. The average gene expression within cells (natural-log scale), fraction of cells expressing a given gene, fold-change between time points (natural-log scale), and false discovery rate (FDR) for differential expression significance are reported. These gene expression, detection, and fold-change calculations are averaged across each of the pairwise comparisons performed, and the max FDR value is shown (see Materials and methods for details on differential expression between time points). Negative fold-change values indicate higher expression at 96 hr and are colored magenta. Positive fold-change values indicate higher expression at 120 hr and are colored green. N.R.: not replicable; calculations in which the fold-change direction differed between pairwise comparisons.

• Supplementary file 4. Geometry of disc model. CSV file of the X, Y, Z geometry used in reference gene expression patterns (*Supplementary file 5*). Formatted as used in DistMap to generate virtual wing disc.

• Supplementary file 5. Reference gene expression patterns. CSV file of the binarized reference gene expression patterns (along with geometry in *Supplementary file 4*). Formatted as used in DistMap to generate virtual wing disc.

• Transparent reporting form

## Data availability

Sequencing data and aligned matrices have deposited in GEO (accession code GSE155543). Code is accessible at https://github.com/HariharanLab/Everetts_Worley_Yasutomi (copy archived at https://archive.softwareheritage.org/swh:1:rev:e1d1f10fefdab11688ad4ca0b8c2684ed47faa0c/). All other data generated are included in the manuscript and supporting files.

The following dataset was generated:

| Author(s) | Year | Dataset title | Dataset URL | Database and Identifier |
|---|---|---|---|---|
| Everetts N, Worley MI, Yasutomi R, Yosef N, Hariharan IK | 2020 | Single-cell transcriptomics of the *Drosophila* wing disc reveals instructive epithelium-to-myoblast interactions | https://www.ncbi.nlm.nih.gov/geo/query/acc.cgi?acc=GSE155543 | NCBI Gene Expression Omnibus, GSE19373 |

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
