## [Decision Letter]

**Acceptance summary:**

This manuscript relates the production of a "cell atlas" of the wing disc and the associated adult muscle precursors using single-cell transcriptomics in *Drosophila* larva. The results of this study are valuable to the community and build well on earlier work by others. The authors propose possible cross-talk between the wing epithelium and the muscle precursors. They identify three ligands that potentially play a role in crosstalk, FGF, Hh, and TNFα. The most relevant results concern the FGF ligands Thisbe and Pyramus, which seem produced by the disc epithelium to regulate the number and the localization of the associated adult muscle precursors.

**Decision letter after peer review:**

Thank you for submitting your article "Single-cell transcriptomics of the *Drosophila* wing disc reveals instructive epithelium-to-myoblast interactions" for consideration by *eLife*. Your article has been reviewed by two peer reviewers, and the evaluation has been overseen by K VijayRaghavan as the Senior and Reviewing Editor. The following individual involved in review of your submission has agreed to reveal their identity: Rajesh D Gunage (Reviewer #2).

The reviewers have discussed the reviews with one another and the Reviewing Editor has drafted this decision to help you prepare a revised submission.

Summary:

This manuscript relates the production of a "cell atlas" of the wing disc and the associated adult muscle precursors using single cell transcriptomics in *Drosophila* larva. While the results of this study are valuable to the community, this is not a novel approach, since two previous studies describe single cell transcriptome analysis for the wing disc and one for the AMPs (Bageritz et al., 2019, Deng et al., 2019, Zappia et al., 2020).

Exploring the expression level of ligand/receptor couples, the authors propose possible cross-talk taking place between the wing epithelium and the AMPs. They identify three ligands that potentially play a role in such crosstalk, FGF, Hh, and TNFα. The most relevant results concern the FGF ligands Thisbe and Pyramus, which seem produced by the disc epithelium to regulate the number and the localisation of the associated AMPs. The proposed role of Hh and Hh signalling is less convincing. No demonstration is made that Hh produced by the disc is indeed signalling to the AMPs. In addition, the phenotype is not sufficiently characterised in the larva. TNF/TNF receptor signalling is not studied.

Overall, the paper presents interesting but spread-out data, due to an substantial focus on the technical aspect of the single cell approach and a general presentation of the data obtained. This approach does not contribute to enhancing interest in the paper and seems dispensable in its details as many parts are essentially similar to previously published approaches. More focus should be put on the real novelty of the study, i.e. FGF/FGFR signalling in the epithelial/muscle crosstalk. As it stands this aspect is somewhat limited.

Essential revisions:

1. The manuscript is 97 pages. This is definitely too long ( the journal does not have a page-limit, but it is prudent to keep a succinct thread going and place details accessible elsewhere). Importantly, the description of the single cell approach and the general exploitation of the data should be moved to supplementary information, given that three previous studies have been published on similar/identical aspects (Figure 1 of this paper is very similar to Figure 1 of Bageritz et al., 2019).

2. The description of FGF signal crosstalk between the disc and the AMPs is potentially interesting, but preliminary. A description of how FGF ligands spread is missing. This is key since over-expression of the ligands seem to trigger localised recruitment of the AMPs. As it stands, the paper does not fully demonstrate that FGF produced by the disc acts on FGF receptor on AMPs. Finally, the distinction between recruitment and proliferation effects is poorly characterised. The notion that FGF signalling allows adapting the number of AMPs to the size of the disc is very interesting. Testing this would be a significant advance, no matter what the result.

3. The same remarks apply to the analysis of the Hh/*ptc*crosstalk. Here a clear characterisation of the larval phenotype is missing. The source of the ligand is also puzzling. Given that Hh is known to accumulate at high level in the hemolymph (Rodenfels et al., 2014) other sources of Hh may also need to be considered. In any case. It may be best, for purposes of this manuscript, that the authors focus on better understanding the FGF pathways here and leave the elaboration of Hh signaling for later.

4. There has been prior work on FGF signaling in mouse ( e.g. Margaret Buckingham's lab) and in the *Drosophila* abdominal muscles. Yet, the elaboration of how ectoderm- mesoderm signalling is effected has not been sufficiently explored in those studies and this study offers such an opportunity, which can be grasped.

---

## [Author Response]

Summary:This manuscript relates the production of a "cell atlas" of the wing disc and the associated adult muscle precursors using single cell transcriptomics in *Drosophila* larva. While the results of this study are valuable to the community, this is not a novel approach, since two previous studies describe single cell transcriptome analysis for the wing disc and one for the AMPs (Bageritz et al., 2019, Deng et al., 2019, Zappia et al., 2020).

We agree that the three studies cited also provided single-cell transcriptomic analysis of the wing disc and that one focused on the AMPs (Zappia et al., 2020). Our work differs from those studies in two important ways. (1) By obtaining data from two different time points during L3, we show that cell fates in the epithelium are established by mid-L3 while AMP fates are consolidated much later. This is key in understanding how signaling from the epithelium impacts the AMPs (2) By performing additional computational analysis of the AMPs (detailed below), we were able to uncover a role for Hedgehog signaling in a subset of the AMPs in specifying fates via AMP-specific target genes that had not been previously described.

We wish we had done a better job of explaining the additional computational analysis that we did that enabled us to find differences between subsets of AMPs that had previously escaped detection. Our analysis showed that Hedgehog signaling is active in a subset of the AMPs – this was not apparent from the data presented in the Zappia et al. paper. This may in part be due to our use of the 10x Genomics barcoding rather than DropSeq technology (which was used extensively by both Bageritz et al. and Zappia et al.) and deep sequencing, which lead to an increased number of genes per cell (an average of ~3,000 genes per cell in our dataset vs. an average of 500 genes per cell in the Zappia et al. dataset). Additionally, as described below, we uniquely addressed confounding biological factors that obscured characterization of the active Hedgehog signaling population within our AMP data.

When the AMPs are analyzed using default parameters, the major stratifying factor that separates the AMPs into four main groups based on both cell sex and cell-cycle status (Please see Figure 1—figure supplements 4 and 5). We found a way of computationally removing stratification from both cell sex and the cell-cycle state in order to focus on cell fates. Stratification based on the cell sex was eliminated by treating male and female cells as distinct batches, which worked well because the relative binary ON/OFF expression of *lncRNA:roX1* and *lncRNA:roX2.* Cell-cycle status is not a binary state. To remove this stratifying effect, we determined which dimension of the latent space effectively captures cell-cycle variation. Briefly, our process of cell cycle correction is a two-step process. In the first step, we use dimensionality reduction to translate the transcriptome of each cell to a 20-dimensional latent space. This step is common among many single-cell RNA analyses, using tools such as PCA or scVI, and is necessary to reduce the noise and complexity of single-cell data for downstream analyses (e.g., clustering for cell-type identification). However, in the second step, we examined each of the 20 latent dimensions for correlation with annotated cell cycle genes. We found that cell-cycle variation was mostly observed in one of these 20 dimensions, and that when we removed this dimension from our analysis (effectively removing cell-cycle effects), we were then able to observe other differences that had been obscured. These two computational corrections allowed us to discover that Hedgehog signaling was active in a particular cluster of AMPs. Importantly, we believe that many of the published datasets from scRNA-seq studies would benefit from such computational corrections.

Once we observed that *patched* was upregulated in a subset of AMPs, we noticed that the canonical target of Hedgehog signaling in the disc epithelium, *decapentaplegic*, was not expressed in these cells. Using computational tools, we looked for genes that were expressed in those cells that upregulated *patched*, and identified *midline* and *Neurotactin* as targets of Hedgehog signaling in AMPs. The discovery of these targets was only possible because of the single-cell approach since we looked for genes that were expressed in the same subset of cells as those that upregulated *patched*.

Thus, while we agree that others have published single-cell studies of the same tissue, the analytical tools and computational corrections that we used enabled us to characterize differences in AMPs with greater resolution than reported in those previous studies, and resulted in the identification and characterization of AMP-specific Hedgehog targets. In addition, by generating an integrated cell atlas from two developmental time points, we were able to investigate the temporal dynamics of gene expression within cell types. This promoted the discovery that the AMPs are still being actively patterned by the epithelium, as observed by the induction of *midline* and *Neurotactin* between mid (96h) and late (120h) phase of wing disc development.

Exploring the expression level of ligand/receptor couples, the authors propose possible cross-talk taking place between the wing epithelium and the AMPs. They identify three ligands that potentially play a role in such crosstalk, FGF, Hh, and TNFα. The most relevant results concern the FGF ligands Thisbe and Pyramus, which seem produced by the disc epithelium to regulate the number and the localisation of the associated AMPs. The proposed role of Hh and Hh signalling is less convincing. No demonstration is made that Hh produced by the disc is indeed signalling to the AMPs. In addition, the phenotype is not sufficiently characterised in the larva. TNF/TNF receptor signalling is not studied.Overall, the paper presents interesting but spread-out data, due to an substantial focus on the technical aspect of the single cell approach and a general presentation of the data obtained. This approach does not contribute to enhancing interest in the paper and seems dispensable in its details as many parts are essentially similar to previously published approaches. More focus should be put on the real novelty of the study, i.e. FGF/FGFR signalling in the epithelial/muscle crosstalk. As it stands this aspect is somewhat limited.

We regret that our most interesting findings were buried in lots of other data. In our revised manuscript, we have presented these more clearly and also included additional experiments that flesh out aspects of FGF and Hedgehog signaling that the reviewer refers to. These findings are detailed below in the section on “Essential revisions”.

Essential revisions:1. The manuscript is 97 pages. This is definitely too long ( the journal does not have a page-limit, but it is prudent to keep a succinct thread going and place details accessible elsewhere). Importantly, the description of the single cell approach and the general exploitation of the data should be moved to supplementary information, given that three previous studies have been published on similar/identical aspects (Figure 1 of this paper is very similar to Figure 1 of Bageritz et al., 2019).

Thank you for pointing this out. We have reduced the word count of the main section of the manuscript by approximately 30% and have reduced the first three figures of the previous version into a single figure. In the text, we emphasize the computational tools that we have used during the course of this work that should be useful to others. As discussed previously, we found that removal of the dominating effects of cell sex and cell cycle on the transcriptome revealed heterogeneity among the AMPs that was not otherwise apparent. We notice that many published studies of multiple cell types in *Drosophila* do not do this and that those data may benefit from this kind of analysis. We have also retained a main figure which describes how we can display our data on a three-layered wing disc (peripodial epithelium, disc proper, and AMPs). This method will enable others to easily visualize the expression of any gene of interest in all three layers of the wing disc.

2. The description of FGF signal crosstalk between the disc and the AMPs is potentially interesting, but preliminary. A description of how FGF ligands spread is missing. This is key since over-expression of the ligands seem to trigger localised recruitment of the AMPs. As it stands, the paper does not fully demonstrate that FGF produced by the disc acts on FGF receptor on AMPs. Finally, the distinction between recruitment and proliferation effects is poorly characterised. The notion that FGF signalling allows adapting the number of AMPs to the size of the disc is very interesting. Testing this would be a significant advance, no matter what the result.

In the revised manuscript, we have included additional data to characterize the role of the FGF ligands in sustaining the viability of the AMPs, in regulating their proliferation, and in recruiting them to alternative locations.

First, we show that the FGF secretion by the epithelial cells is necessary for the survival of the AMPs in the notum (Figure 3H). We examined cell death using an antibody that recognizes activated caspase (anti-Dcp1). Following knockdown of *pyramus*, we observe elevated levels of caspase activity in the same parts of notum where myoblast numbers are decreased, especially in the ventral part of the notum. This is the region where *pyramus* and *thisbe* are not co-expressed, as evidenced by spatial mapping of our single-cell data, and therefore unlikely to function redundantly. This also suggests that at physiological levels of expression, these FGFs are acting as short-range signals, since there is insufficient Thisbe in this region to compensate for reduced Pyramus levels.

Second, we show (in multiple ways) that the FGF ligands can promote the proliferation of AMPs. We had previously shown that while antagonizing FGF signaling reduced AMP numbers (which could simply be a pro-survival effect), increasing FGF levels using *dpp-Gal4* caused a spectacular increase in the number of AMPs. Our previous experiments could not distinguish if these cells were proliferating only in their normal niche (in the notum) and then emigrating or whether they were proliferating in these ectopic locations. We have added data that show that AMPs that are far from the notum continue to proliferate (Figure 4E, F; Figure 4—figure supplement 1D, E). We detect ectopic AMPs undergoing both S-phase and mitosis by staining for EdU incorporation and anti-phosphohistone H3, respectively. Thus, we now show that AMPs can proliferate at many locations in the wing disc as long as sufficient levels of FGF ligands are provided. Taken together, these experiments show that the FGF ligands are necessary for the survival of AMPs and that altering their levels can regulate AMP numbers.

Third, we have examined the time course of AMP migration out of the notum when ectopic FGF is provided. Our previous experiments did not address the issue of when AMPs were able to be drawn out of the notum. Using *dpp-Gal4* together with a temperature-sensitive *Gal80*, we initiate expression of *>pyr* and *>ths* at different stages of larval development (Figure 4G, H; Figure 4—figure supplement 1B, C). We show that even when expression is initiated at mid L3, we can still induce migration of AMPs from below the notum epithelium to new locations beneath the ventral hinge and pouch. This shows that the FGFs do not function simply in the initial localization of the AMPs to the notum region of the disc and argues that the AMPs remain in this region because it is the only portion of the epithelium that continues to expresses the FGF ligands *ths* and *pyr*.

Fourth, our experiments have also addressed the issue of how far FGF ligands can spread. Under conditions of overexpression, we show that sufficient FGF can diffuse many cell diameters from the source to enable AMPs to emigrate to and survive at those locations. However, at physiological levels of expression, we do not observe AMPs far from the epithelial cells that make them, indicating that insufficient FGF reaches regions of the epithelium outside of the notum to draw AMPs to those locations.

In summary, the data we have added make a strong argument that localized expression of the FGF ligands is necessary for the survival of the AMPs, for regulating the extent of their proliferation, and for restricting them to the region of the notum.

We also attempted to visualize the FGF ligands in the AMPs. We have spent the last two months trying to visualize the spread of tagged versions of the FGF ligands. These stainings have been occasionally successful and in Author response image 1, we observe endogenous Thisbe-FLAG (visualized with anti-FLAG) in the epithelial cells of the notum and in puncta in the underlying AMPs. We have tried a variety of fixation conditions and detergents but found the staining to be inconsistent and with varying amounts of background. For this reason, we are reluctant to include these images in the main manuscript. Importantly, in the images shown, the puncta we observe remain confined in myoblasts under the Ths-expressing part of the notum and they do not extend more ventrally even to the region that are predicted from our single-cell data to express *pyramus*. While these images are not pretty, the result is consistent with our other experiments that show that at physiological levels of expression the FGF ligands act at short distances from their source.

**Author response image 1. sa2fig1:** FLAG-tagged Ths ligand is observed in both the epithelium notum and underlying AMPs. (A-C) Max projections over all images slices (epithelium + AMPs) from wing discs with FLAG tag inserted at the N-terminus of Ths (Bl. 77476). Single image slice of the disc proper (DP) notum (D-F) or underlying AMPs (G-I), corresponding to the region within the yellow dashed box in A. (J-L) Close up of AMPs within the dashed yellow box in G. In all images, FLAG is visualized with anti-FLAG antibody (red), and AMPs are visualized with anti-Cut antibody (cyan). Anti-Cut antibody imperfectly stains AMPs, due to the use of saponin to optimize the visualization of anti-FLAG staining. Note the puncta stained by anti-FLAG antibody within the AMPs. Microscopy scale bars = 100 μm in A-I, and 10 μm in J-L.

3. The same remarks apply to the analysis of the Hh/ptc crosstalk. Here a clear characterisation of the larval phenotype is missing.

Our Hedgehog work was presented at the end of a long manuscript and the reviewer may have overlooked our data on the characterization of the larval phenotype. We had devoted an entire figure (and many experiments) to characterizing the larval phenotype (now Figure 6). Indeed, we feel that discovering AMP-specific targets of Hedgehog signaling in the larval wing disc best illustrates the power of single-cell transcriptomics.

We found that *dpp* was not expressed by the *ptc*-expressing AMPs. We leveraged our single cell data to show that the transcription factor Midline and the cell-surface protein *Neurotactin* are expressed in cells that upregulate *patched* (Looking for co-expression in subsets of cells is one of the most powerful uses of single-cell transcriptomics). We showed that the expression of both Midline and *Neurotactin* is dependent on Hedgehog signaling. Thus, we have identified AMP-specific targets of Hh in the larval disc.

Our work also suggests that AMPs are displaced anteriorly during larval development since we detect perdurance of Midline and *Neurotactin* in more anteriorly located AMPs and present lineage-tracing data to show that these cells were once located more posteriorly.

The source of the ligand is also puzzling. Given that Hh is known to accumulate at high level in the hemolymph (Rodenfels et al., 2014) other sources of Hh may also need to be considered.

We are aware of the work from Rodenfels that Hh can accumulate in the hemolymph. However, we have now added a series of images (Figure 5—figure supplement 1) that show that the only AMPs that upregulate *patched* are those that are within approximately 40 μm of the cells in the Hedgehog-secreting cells in the posterior compartment of the disc epithelium. This makes a strong argument that Hedgehog is not freely available to all AMPs.

In any case. It may be best, for purposes of this manuscript, that the authors focus on better understanding the FGF pathways here and leave the elaboration of Hh signaling for later.

We believe that the revised manuscript has strengthened both the FGF and Hedgehog sections of the paper.

4. There has been prior work on FGF signaling in mouse ( e.g. Margaret Buckingham's lab) and in the *Drosophila* abdominal muscles. Yet, the elaboration of how ectoderm- mesoderm signalling is effected has not been sufficiently explored in those studies and this study offers such an opportunity, which can be grasped.

Work in the Buckingham laboratory has demonstrated autocrine FGF signaling in the precursors of the cardiac outflow tract (mesodermal cells). The work with the adult *Drosophila* abdominal muscles has shown a role for FGF signaling in selecting founder cells but does not seem to be involved in regulating the overall number of myoblasts.

In the wing disc, we have shown that the two FGF proteins function to organize a portion of the epithelium as a myoblast niche. We have shown that (1) their expression is necessary for AMP survival, (2) their levels can regulate myoblast number by promoting AMP proliferation, (3) that AMPs can migrate towards a source of FGF, and (4) that secretion of FGF ligands at many different locations in the disc epithelium is sufficient to sustain myoblast proliferation at those locations. Thus, we have shown that the FGF-secreting epithelial cells constitute the AMP niche.